# Contributions of cortical neuron firing patterns, synaptic connectivity, and plasticity to task performance

Michele N. Insanally [1,2,3,4,18] ✉, Badr F. Albanna [1,18], Jade Toth[1,2], Brian DePasquale[5,6], Saba Shokat Fadaei[7,8,9,10,11], Trisha Gupta[1,2], Olivia Lombardi[1,2], Kishore Kuchibhotla [12,13,14], Kanaka Rajan [15,16] & Robert C. Froemke [7,8,9,10,11,17] ✉

Neuronal responses during behavior are diverse, ranging from highly reliable 'classical' responses to irregular 'non-classically responsive' firing. While a continuum of response properties is observed across neural systems, little is known about the synaptic origins and contributions of diverse responses to network function, perception, and behavior. To capture the heterogeneous responses measured from auditory cortex of rodents performing a frequency recognition task, we use a novel task-performing spiking recurrent neural network incorporating spike-timing-dependent plasticity. Reliable and irregular units contribute differentially to task performance via output and recurrent connections, respectively. Excitatory plasticity shifts the response distribution while inhibition constrains its diversity. Together both improve task performance with full network engagement. The same local patterns of synaptic inputs predict spiking response properties of network units and auditory cortical neurons from in vivo whole-cell recordings during behavior. Thus, diverse neural responses contribute to network function and emerge from synaptic plasticity rules.

Neuronal spiking patterns and responses to sensory input can be remarkably diverse, ranging from completely silent or firing a single action potential to prolonged burst firing or complex sequence generation[1]. Various spiking patterns have been documented throughout brain regions in response to different sensory inputs, in relation to decision making, motor actions, or other task-related signals[2–31]. The extent of spiking and receptive field heterogeneity is vast, with numerous types of neuronal responses found in many brain

[1]Department of Otolaryngology, University of Pittsburgh School of Medicine, Pittsburgh, PA 15213, USA. [2]Pittsburgh Hearing Research Center, University of Pittsburgh, Pittsburgh, PA 15213, USA. [3]Department of Neurobiology, University of Pittsburgh School of Medicine, Pittsburgh, PA 15213, USA. [4]Department of Bioengineering, University of Pittsburgh, Pittsburgh, PA 15213, USA. [5]Department of Biomedical Engineering, Boston University, Boston, MA 02215, USA. [6]Center for Systems Neuroscience, Boston University, Boston, MA 02215, USA. [7]Skirball Institute for Biomolecular Medicine, New York University Grossman School of Medicine, New York, NY 10016, USA. [8]Neuroscience Institute, New York University Grossman School of Medicine, New York, NY 10016, USA. [9]Department of Otolaryngology, New York University Grossman School of Medicine, New York, NY 10016, USA. [10]Department of Neuroscience, New York University Grossman School of Medicine, New York, NY 10016, USA. [11]Department of Physiology, New York University Grossman School of Medicine, New York, NY 10016, USA. [12]Department of Psychological and Brain Sciences, Johns Hopkins University, Baltimore, MD 21218, USA. [13]Department of Neuroscience, Johns Hopkins University, Baltimore, MD 21218, USA. [14]Department of Biomedical Engineering, Johns Hopkins University, Baltimore, MD 21218, USA. [15]Department of Neurobiology, Harvard Medical School, Boston, MA 02115, USA. [16]Kempner Institute, Harvard University, Cambridge, MA 02138, USA. [17]Center for Neural Science, New York University, New York, NY 10003, USA. [18]These authors contributed equally: Michele N. Insanally, Badr F. Albanna. ✉e-mail: mni@pitt.edu; robert.froemke@med.nyu.edu

areas including visual cortex[2–5], auditory cortex[6–12], somatosensory cortex[13], parietal cortex[4,14,15], frontal cortex[15–19], hypothalamus[20–22], hippocampus[23–26], and the ventral tegmental area[27,28] correlated with sensory, motor, choice, and other task-related signals[2,4,11,14–16,29,30]. These neurons are often described as tuned, untuned, classically responsive, non-classically responsive, mixed selective, or category-free[14,16,18,31].

"Classically responsive neurons" (or "strongly rate-modulated neurons") are those that have clear trial-averaged evoked activity relative to baseline in response to stimulus or other task-related events. "Non-classically responsive neurons" (or "weakly rate-modulated neurons") are the remaining neurons that do not have these features and appear non-responsive. As shown previously[16], despite little to no rate modulation, non-classically responsive neurons encode substantial information about stimulus or choice behavioral variables in the relative timing of their spikes. Thus, it is important to determine whether these neurons contribute to the computations underlying task performance.

Recently, we showed that classically responsive neurons (e.g., pure tone frequency tuning in auditory cortex) and non-classically responsive neurons (i.e., nominally "non-responsive" neurons) both contained significant information about sensory stimuli and behavioral decisions. This finding suggests that non-classically responsive cells play important yet generally underappreciated roles in perception and behavior[16]. This work is consistent with other recent findings demonstrating that neurons in rat primary visual and parietal cortex can encode sensory and non-sensory factors related to movement, reward history, and decision making[4]. Similarly, a recent study on sequential memory in humans found that both strongly tuned and weakly tuned neurons recorded from the medial temporal lobe participated in theta-phase-locked encoding of sequence stimuli[32]. A previous study on working memory in primate prefrontal cortex also revealed that non-selective neurons can contribute to optimal ensemble encoding[19], consistent with our own finding that mixed ensembles of classically and non-classically responsive cells improved encoding of task variables[16]. Studies of deep neural networks trained to perform a visual recognition task showed that regularizing networks to increase the fraction of 'non-selective' units improved network performance relative to those with greater numbers of 'selective' units[33]. These findings suggest that a diversity of neuronal response types (including neurons nominally thought to be non-responsive) may be a general property of neural networks, and that heterogeneity may be a key feature of the circuit dynamics important for network performance and behavior.

Here we now examine the synaptic basis for various types of spiking response profiles, and how synaptic plasticity learning rules were important for shaping synaptic inputs for spike output and network performance. We leveraged cell-attached, extracellular, and whole-cell recordings from behaving animals alongside recurrent network modeling to explore the synaptic origins and functional contribution of heterogeneous response profiles. Recent advancements using recurrent neural networks (RNN) with spiking units have shown that experimentally derived synaptic plasticity mechanisms can support stable neuronal assemblies[34] and coordinate memory formation and retrieval[35]. Related work has also shown that spiking RNNs can be trained to perform tasks using general-purpose methods similar to those employed in rate-based networks such as first-order reduced and controlled error (FORCE) training[36,37] but these methods have only been employed as perturbation of networks with static synaptic weights. We combined FORCE training with a dynamic network to create a novel RNN with spiking units and multiple spike-timing-dependent plasticity (STDP) rules to solve a stimulus classification task similar to that of trained rats and mice. Our goal was to determine whether and how classically and non-classically responsive units contribute to task performance, how local synaptic structure (e.g.

monosynaptic and disynaptic connections) constrains single-unit response profiles, and if the relationships observed for units in our network model could be applied to neurons in vivo during behavior.

## Results

### Diverse cortical responses measured during behavior in freely moving rats and head-fixed mice

We recorded from the rodent auditory cortex as animals performed a task requiring them to classify specific tone frequencies. We first trained rats to perform a go/no-go auditory frequency recognition task (Fig. 1a) requiring them to behaviorally respond with a nosepoke to a single target tone (4 kHz) for food reward and to withhold responses on non-target tones (0.5, 1, 2, 8, 16, 32 kHz). Rats learned this task within a few weeks of training performing with high d' values (Fig. 1b; $d'$ = 2.8 ± 0.1, $p$ = 6.1 × 10$^{-5}$, $N$ = 15, two-sided Wilcoxon test). Perturbation experiments in rodents have revealed that the contribution of auditory cortex to perceptual decision making depends on details of the task design and difficulty[38]; however, we and others have previously shown using muscimol inactivation studies that the auditory cortex is required for this task[7,16,39]. (Although it is important to keep in mind that many loss of function experiments may be challenging to appropriately control for and interpret[38].) After rats reached behavioral criteria (percent correct: ≥70%, $d'$: ≥1.5), tetrodes were implanted in the right auditory cortex and we recorded from populations of single-units in non-head-fixed animals as they performed this go/no-go task (Fig. 1c). Single-unit responses were quite diverse across the population, spanning a range of response types from 'classically responsive' cells that were highly modulated relative to pre-trial baselines during the task to 'non-classically responsive' cells with relatively unmodulated firing rates throughout task performance including cue presentation and behavioral choice.

To capture the continuum of response types, we calculated a 'firing rate modulation index' comparing neural responses during the stimulus and choice periods to baseline values where either positive or negative changes in spike number increase the modulation index (in units of spikes per second). A low value of firing rate modulation index (near 0 spikes/s) corresponds to neurons that were unmodulated relative to baseline ('non-classically responsive') and larger values (≥2 spikes/s) correspond to neurons that were highly modulated ('classically responsive'). The modulation was calculated from the firing rates during the stimulus and choice periods ('stimulus/choice FR') and baseline firing rate ('baseline FR') as

$$\text{modulation} = \sqrt{(\text{stimulus FR} - \text{baseline FR})^2 + (\text{choice FR} - \text{baseline FR})^2}$$

(1)

Many single-units recorded during behavior were non-classically responsive, with the remainder of cells having more 'classical' responses, e.g., to tone presentation (Fig. 1d, e; median firing rate modulation = 0.78 spikes/s, interquartile range = 0.47–1.50 spikes/s).

We observed a similar range of neuronal response profiles in cell-attached recordings from the auditory cortex of head-fixed mice trained on an analogous go/no-go auditory frequency recognition task (Fig. 1f). Mice were trained to respond to a single target tone (11.2 kHz) by licking for a water reward, and to withhold their response to a single non-target tone (5.6 kHz). Mice learned this task within a few weeks performing at high d' values (Fig. 1g; $d'$ = 2.5 ± 0.1, $p$ = 0.016, $N$ = 7 mice, two-sided Wilcoxon test). After reaching behavioral criteria (percent correct ≥70% and $d'$: ≥1.5), animals were implanted with a cranial window that included a small hole for pipette access, and cell-attached recordings were made to measure spike firing during behavior (Fig. 1h). We found that neuronal responses in mice were also heterogeneous, including non-classically responsive cells with low firing rate modulation as well as classically responsive cells that were highly modulated during the task (Fig. 1i, j; median firing rate modulation =

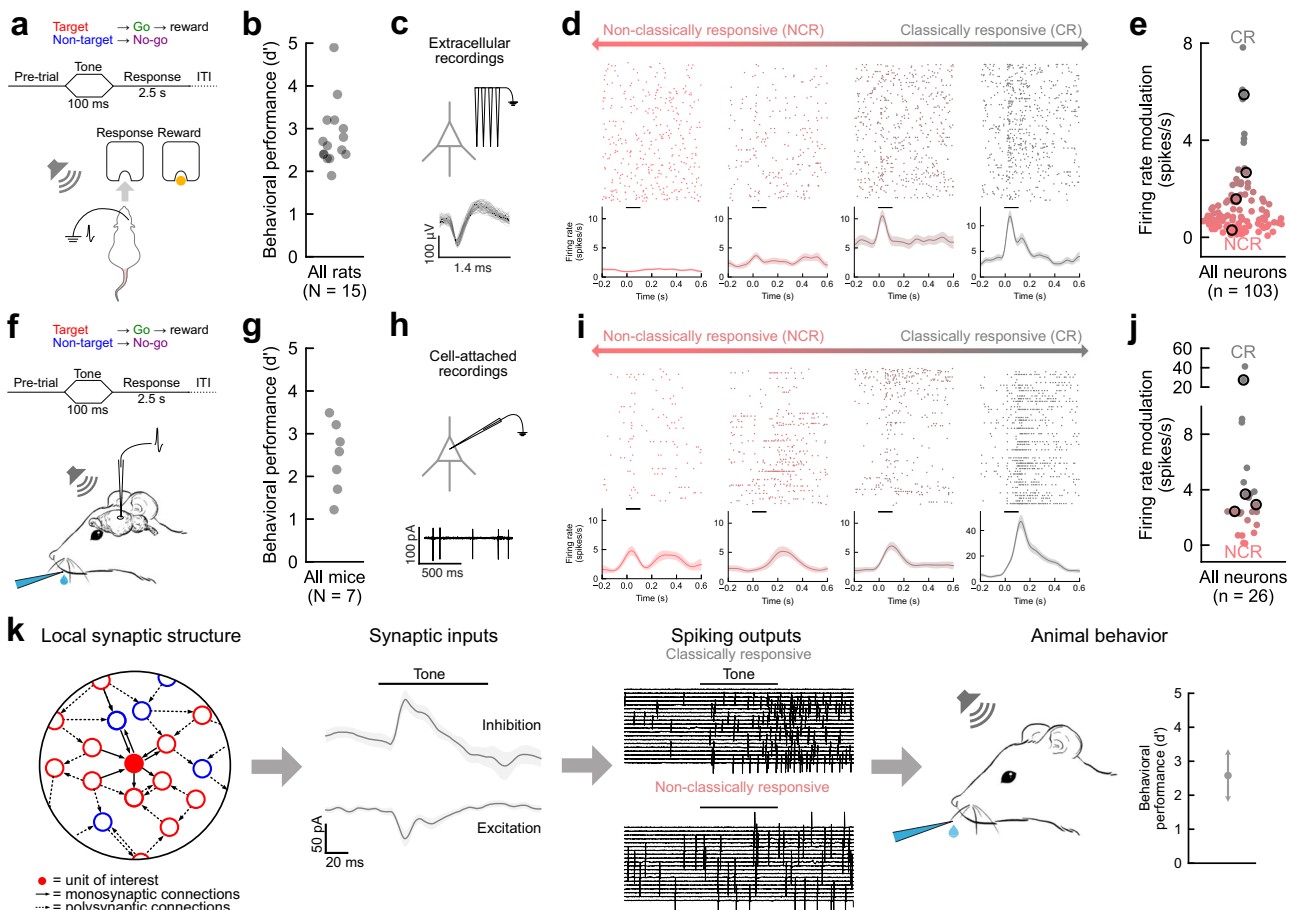

**Fig. 1 | Diverse single-unit responses measured in rodent auditory cortex during behavior. a** Schematic of behavior and extracellular tetrode recordings from auditory cortex of rat performing go/no-go frequency recognition task. **b** Asymptotic behavioral performance for all rats ($d' = 2.8 \pm 0.1$, $p = 6.1 \times 10^{-5}$, $N = 11$ rats, two-sided Wilcoxon test). Source data are provided as a Source Data file. **c** Example single-unit recording from rat auditory cortex during behavior. **d** Rasters and peri-stimulus time histograms (PSTHs) for four cortical neurons exemplifying the range from non-classically responsive (red, NCR) to classically responsive (gray, CR). Lines in PSTH, mean firing rate; error bands, SEM; horizontal bar, tone duration. **e** Summary of firing rate modulation for all cortical neurons recorded during behavior ($n = 103$). Outlined circles, units from (**d**). Median firing rate modulation = 0.78 spikes/s (inter-quartile range 0.47–1.50 spikes/s). Source data are provided as

a Source Data file. **f** Cell-attached recordings from auditory cortex of mouse performing go/no-go frequency recognition task. **g** Asymptotic behavioral performance for all mice ($d' = 2.45 \pm 0.11$, $p = 0.016$, $N = 7$ mice, two-sided Wilcoxon test). Source data are provided as a Source Data file. **h** Example cell-attached recording from mouse auditory cortex during behavior. **i** Rasters and PSTHs for four example recordings. Lines in PSTH, mean firing rate; error bands, SEM; horizontal bar, tone duration. **j** Firing rate modulation for all cell-attached recordings ($n = 26$ cells) from mouse auditory cortex during behavior (median firing rate modulation = 2.26, interquartile range = 1.73–3.61 spikes/s). Source data are provided as a Source Data file. **k** Diagram of relationship between local synaptic structure, synaptic inputs, spiking outputs, and behavior. Artwork in (**f**, **k**) by Shari E. Ross.

2.26, interquartile range = 1.73–3.61 spikes/s). In both rats and mice, baseline firing rates for non-classically responsive neurons were comparable or lower than those of responsive neurons (Supplementary Fig. 1). Apparent differences in the response distribution between rats and mice may be attributable to species or task differences, but in both cases a wide range of response profiles were observed. These data then led us to wonder how local patterns of single-neuron excitatory and inhibitory inputs related to spike firing and behavioral performance (Fig. 1k).

### A spiking RNN model incorporating STDP rules captures in vivo cortical dynamics

To relate inputs and outputs over the response-type continuum, we developed a spiking RNN model trained to perform a similar go/no-go stimulus classification task as behaving animals (Fig. 2a–c; $d' = 4.6 \pm 0.1$, $p = 0.0078$, $N = 8$ networks, two-sided Wilcoxon test). All networks contained 1000 units (200 inhibitory, 800 excitatory); 200 excitatory units received inputs (input units) and the remaining 600 excitatory units projected onto the readout node (output units). The activity of

this node is the dynamic signal which represents the response of our network to the incoming stimuli: 'go' is represented by an increase in node activity during the choice period, and 'no-go' is represented by node activity that remains at pre-trial baseline. All units were recurrently connected with a 5% random connection probability. The network was trained to perform the task via a version of FORCE designed for spiking networks[36,37] (i.e., least-squares modification of the output weights with feedback) combined with STDP synaptic plasticity learning rules acting on the recurrent weights (Fig. 2a, Supplementary Fig. 2a–d). The network was trained to use a minimal amount of external stimulus input while being able to perform the task to mirror the behavioral errors seen during animal performance (Supplementary Fig. 2e).

Our model included biologically motivated and experimentally constrained excitatory and inhibitory synaptic STDP. Excitatory-to-excitatory synapses were modified by classic pairwise Hebbian homosynaptic plasticity[40,41], and inhibitory-to-excitatory synapses were modified by a homosynaptic rule which strengthens synapses when units fire synchronously regardless of order[42–45]. Excitatory-to-

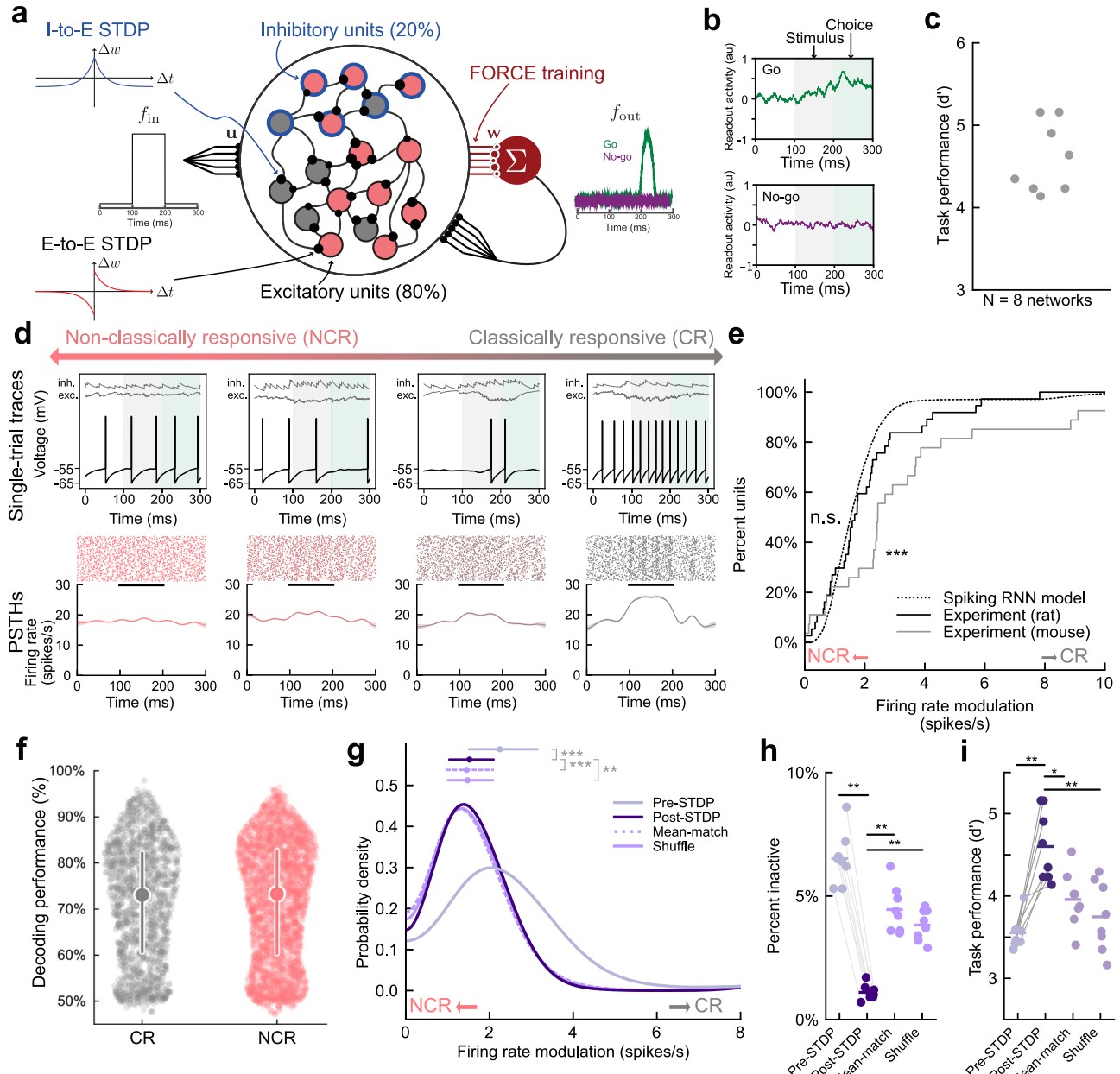

**Fig. 2 | A spiking RNN model incorporating STDP rules can recapitulate in vivo cortical neuronal dynamics. a** Schematic of spiking RNN model trained to complete go/no-go stimulus recognition task. 80% excitatory (800 units) and 20% inhibitory units (200 units). 25% of excitatory units (200 units) received direct current as stimulus (1 target, 6 non-target) and remaining 75% were output units (600 units) projecting to and receiving feedback from readout node (maroon). Weights to readout node trained via FORCE. Excitatory-to-excitatory and inhibitory-to-excitatory synapses modified by distinct STDP mechanisms. **b** Example readout node activity on 'go' trial (in response to 'target') and 'no-go' trial (in response to 'non-target'). White, pre-trial baseline; gray, stimulus period; green, choice period. **c** Asymptotic task performance. $d'$ = 4.6 ± 0.1, $p$ = 0.0078, $N$ = 8 networks, two-sided Wilcoxon test. **d** Top, example single-trial voltage traces spanning non-classically to classically responsive units. Bottom, example spike rasters across trials and PSTHs. **e** Cumulative distribution of single-unit firing rate modulation for spiking RNN model (dotted) and experimental data (rat: solid black, mouse: solid grey). Simulated vs. rat were not statistically distinct ($p$ = 0.27, two-sided Kolmogorov-Smirnov test) simulated vs. mouse were distinct ($p$ = 1.46 × 10⁻⁷,

two-sided Kolmogorov–Smirnov test). **f** Single-unit decoding performance ($N$ = 1 network, $n$ = 1000 units) for classically (CR, grey, left) and non-classically responsive units (NCR, red, right). Circle and line represent median and interquartile range. **g** Probability density of single-unit firing rate modulation. Gray, pre-STDP; purple, post-STDP; dotted light purple, mean-matched control; solid light purple, shuffle control. Circles and bars above distributions represent median and interquartile range. Median post-STDP modulation = 1.52 vs. pre-SDTP modulation = 2.25, $p$ = 1×10⁻³²³, vs. mean-match modulation=1.46, $p$ = 6.2 × 10⁻⁶, vs. shuffle modulation = 1.48, $p$ = 0.0017, $N$ = 8 networks in all groups, two-sided Mann−Whitney $U$ test with Bonferroni correction. **h** Percent inactive units for pre-STDP, post-STDP, and controls defined as firing rate <1 spikes/s. All comparisons to post-STDP, $p$ < 0.001, $N$ = 8 networks in all groups, two-sided Mann−Whitney $U$ test with Bonferroni correction. **i** Task performance for pre-STDP, post-STDP, and controls. Mean shifts relative to pre-STDP, post-STDP: Δ$d'$ = 0.97, $p$ = 0.0014, mean-match: Δ$d'$=0.44, $p$ = 0.04, shuffle: Δ$d'$ = 0.15, $p$ = 0.74, $N$ = 8 networks in all groups, two-sided Wilcoxon test with Bonferroni correction. Source data are provided as a Source Data file.

excitatory and inhibitory-to-excitatory synapses were also adjusted by homeostatic mechanisms of heterosynaptic plasticity which prevented any one presynaptic connection from dominating (heterosynaptic balancing, β) or postsynaptic connection from dropping out (heterosynaptic enhancement, δ). These heterosynaptic changes occurred simultaneously with homosynaptic mechanisms and thus were qualitatively distinct from other types of homeostatic regulation of input weight distributions such as synaptic scaling.

While FORCE was originally designed to construct networks that can solve tasks without modification of the underlying recurrent synapses, in our network, STDP constrains FORCE to find solutions consistent with biological plasticity rules. FORCE and STDP can operate in parallel because each mechanism can be targeted to a different set of connections in the model: STDP modifies the recurrent synapses while FORCE modifies the connections to the readout node. In this way, STDP directly shaped the inherent recurrent dynamics of our network while FORCE determined how those dynamics were harnessed to perform the task. Since FORCE was originally designed to operate in networks with fixed recurrent weights and in our network these evolve, gross changes induced by STDP must be complete before FORCE can adapt to smaller synaptic changes. To ensure this, STDP was first activated without FORCE to allow for initial major synaptic restructuring to occur before FORCE training began and the two mechanisms continued in parallel (see Methods). The particular mechanisms and parameters chosen for STDP in this model were not optimized for the task a priori, and as such STDP should not have trivially improved performance. In general, we found that by using this procedure FORCE was compatible with STDP over a wide range of STDP parameters (Supplementary Fig. 2f–h). All networks without STDP active were trained with FORCE for the same number of trials as those with STDP.

We found that the spiking responses of individual network units spanned a wide range from non-classically to classically responsive similar to the in vivo data from both rats and mice (Fig. 2d, e; Fig. 1d, e, i, j). Specifically, spiking networks with STDP closely approximated the distribution of firing rate modulations observed experimentally in the rat auditory cortex in vivo while they were systematically lower than the modulation observed in mice (Fig. 2e, Supplementary Fig. 3a; for RNN vs. rat experimental data $p = 0.27$, two-sided Kolmogorov-Smirnov test, for RNN vs. mouse experimental data $p = 1.46 \times 10^{-7}$, two-sided Kolmogorov–Smirnov test). Using a statistical threshold to identify non-classically responsive units as previously described[16] (firing rate change from baseline <0.2 spikes/s during stimulus and choice periods, see Methods), we found the relative fractions of classically and non-classically responsive units were also comparable to experimental measurements (Supplementary Fig. 3b–d; ~40–50% non-classically responsive, 50-60% classically responsive). Using a single-trial, interspike interval (ISI)-based, Bayesian decoder we recently described[16], we found that task information was encoded in the activity of both classically and non-classically responsive RNN units (Fig. 2f).

To assess whether STDP altered response profile distributions, we compared the distribution of units before STDP was applied (pre-STDP) to those after (post-STDP). In pre-STDP networks, the recurrent weights were fixed during FORCE training whereas in post-STDP networks the recurrent weights evolved according to the STDP rules described above. Pre- and post-STDP networks were constructed in pairs with the same set of initial recurrent weights so that pre-STDP networks represent the behavior of the network with FORCE alone before STDP was active. The post-STDP response profile distribution differed substantially from pre-STDP networks, such that post-STDP networks exhibited more non-classically responsive units than pre-STDP networks (Fig. 2g; median post-STDP modulation = 1.52 spikes/s vs. pre-STDP modulation = 2.25 spikes/s, $p = 1 \times 10^{-323}$, two-sided Mann–Whitney $U$ test with Bonferroni correction). To understand how the synaptic changes induced by STDP resulted in this shift, we created

two control networks with engineered weight matrices based on the full post-STDP weights. To determine whether the post-STDP response profile distribution resulted from a shift in mean weight strength, we first created networks with weights generated as in the pre-STDP condition but with mean values matched to those found post-STDP (Fig. 2g; 'Mean-match', weights drawn from a uniform distribution from 0 to 2x mean, median post-STDP modulation = 1.52 vs. mean-match modulation = 1.46, $p = 6.2 \times 10^{-6}$, $N = 8$ networks; Supplementary Fig. 4a). Second, we created networks in which inhibitory-to-excitatory and excitatory-to-excitatory synaptic weights were shuffled at random to new pre- and postsynaptic target units to preserve the full weight distribution created by STDP but remove any synaptic correlations (Fig. 2g, 'Shuffle', median post-STDP modulation = 1.52 vs. shuffle modulation = 1.48, $p = 0.0017$, $N = 8$ networks, Supplementary Fig. 4a). Both control conditions were retrained with FORCE after weight matrices were altered. For both the 'mean-match' and 'shuffle' controls, the distributions of responses closely followed the post-STDP distribution; however, the proportion of inactive units (firing rate <1 spike/trial on average) significantly increased to ~5% of the full network (Fig. 2h; comparisons to post-STDP, $p = 0.001$, two-sided Mann–Whitney $U$ test with Bonferroni correction). These results show that the overall synaptic weight changes induced by STDP could account for the increase in the fraction of non-classically responsive units. However, the synaptic correlations induced by STDP were required to facilitate full network engagement (i.e., no inactive units).

To determine how STDP shaped synaptic weight strengths and structure (i.e., network topology), we compared the synaptic weight strengths pre-STDP to post-STDP. Over the course of training, we found that STDP mechanisms made significant modifications to both the excitatory-to-excitatory and inhibitory-to-excitatory synaptic weight distributions and topologies. Median weights shifted relative to pre-STDP initial values for both types of synapses and the distributions of synaptic weights became skewed (Supplementary Fig. 4b), as has been observed in rat visual cortex[46]. We asked if selective weight rescaling by STDP preserved the random structure of the initial pre-STDP network or created systematic patterns of synaptic connectivity (i.e., "small-world" network topology) as previously described in rat somatosensory cortex[47,48]. We compared the topology of the synaptic weight matrix post- and pre-STDP and found that STDP generated a more clustered network structure consistent with both experimental observations[48] and computational studies[34,35] (Supplementary Fig. 4c).

Although our network was not explicitly designed to model multiple regions, we investigated whether our STDP rules induced synaptic structure between the input units (which receive the stimuli) and the output units (which determine the network response). We found that the STDP mechanisms in our model generate stimulus-specific correlations between excitatory and inhibitory inputs to output units consistent with feed-forward inhibition between thalamus and auditory cortex responsible for lateral refinement of sensory representations[49] (Supplementary Fig. 4d). Future work exploring multi-region RNN architectures is required to determine whether these STDP rules are sufficient to reproduce more detailed aspects of the interaction between auditory cortex and upstream or downstream regions.

The modifications to the weight matrix made by STDP improved task performance (Fig. 2i; mean pre-SDTP $d' = 3.52$ vs. post-STDP $d' = 4.49$, $p = 0.0014$, $N = 8$ networks, two-sided Wilcoxon test with Bonferroni correction). This increase in performance was not observed in either the mean-matched or shuffled control conditions, demonstrating that the detailed connectivity structure of synaptic weights created by STDP were required to improve performance (Fig. 2i; mean pre-STDP $d' = 3.52$ vs. mean-match $d' = 3.96$, $p = 0.04$, and shuffle $d' = 3.67$, $p = 0.74$, $N = 8$ networks in all groups, two-sided Wilcoxon test with Bonferroni correction). These results show that STDP shifts the weight matrix into a regime leading to improved task performance while maintaining a consistent level of network unit engagement.

## Classically and non-classically responsive RNN units differentially contribute to task performance

Given that STDP increased both the prevalence of non-classically responsive units as well as network task performance, we next explored how classically responsive and non-classically responsive units contributed directly to task performance. Given that excitatory output units contribute directly to the RNN output activity and recurrent dynamics, we focused our analysis on these units ($n = 600$ per network). We first evaluated the output connections to the readout node and recurrent weights between units to determine whether there were systematic differences between these response profiles. Both classically and non-classically responsive units spanned a similar range of values, however classically responsive units had larger weights onto the readout node than non-classically responsive units (Fig. 3a; $p = 4.9 \times 10^{-22}$, $N = 8$ networks, $n = 4800$ connections, Levene's test). This suggests that both classes contribute directly to task performance by driving the network output, but classically responsive units may affect performance specifically via their effect on the readout node. In contrast, non-classically responsive units had stronger recurrent projections than classically responsive units to both subpopulations (Fig. 3b; $p < 2.0 \times 10^{-5}$, $N = 8$ networks, $n = 4800$ units, for all comparisons between NCR and CR, two-sided Hierarchical Bootstrapping Test). This result, coupled with the observation that non-classically responsive units generally had higher baseline and full-trial firing rates (Fig. 3c, median baseline NCR = 16.1 spikes/s vs. CR = 14.1 spikes/s, $p < 2.0 \times 10^{-5}$; full-trial NCR = 16.1 spikes/s vs. CR = 14.3 spikes/s, $p < 2.0 \times 10^{-5}$, $N = 8$ networks, $n = 4800$ units, two-sided Hierarchical Bootstrapping Test) suggests non-classically responsive units may play a privileged role in generating task-related dynamics through their effect on recurrent network activity. Notably, in vivo average baseline firing rates for non-classically responsive neurons were lower than classically responsive neurons in both rats and mice (Supplementary Fig. 1) this difference may result from the fact that default network parameters prevented neurons with very low average firing rates from emerging.

To test this hypothesis, we transiently inactivated classically or non-classically responsive units during task performance. Inactivation of non-classically responsive units targeted the least modulated units in the network first while inactivation of classically responsive units targeted the most modulated units. During inactivation, we silenced output connections and replaced recurrent activity with average-firing-rate-matched Poisson noise to control for changes in the overall level of recurrent synaptic current. Completely inactivating either subpopulation impaired task performance suggesting that both subpopulations are important for network dynamics. Inactivating a relatively small number of highly non-classically responsive units (60 units, top 10% most non-classically responsive output units) had a significantly larger effect on performance than inactivating classically responsive units (Fig. 3d, Supplementary Fig. 5a; inactivating 10% most non-classically responsive $\Delta d' = -1.58$ vs. inactivating 10% most classically responsive $\Delta d' = -0.92$, $p = 1.11 \times 10^{-4}$, $N = 24$ networks, two-sided Mann–Whitney $U$ test with Benjamini–Hochberg correction). As the number of inactivated units increased, however, task performance continued to degrade and eventually the effects of inactivating 300 units (i.e., 50% of output units) from either subpopulation were comparable (Fig. 3d; inactivating 50% most non-classically responsive $\Delta d' = -2.27$ vs. inactivating 50% most classically responsive $\Delta d' = -2.61$, $p = 0.042$, $N = 24$ networks, two-sided Mann–Whitney $U$ test with Benjamini–Hochberg correction).

The types of errors produced ('false alarms' on non-target trials and 'misses' on target trials) did not differ when inactivating either classically or non-classically responsive units (Supplementary Fig. 3c; false alarms vs. misses inactivating 50% most classically responsive, $p = 0.26$, inactivating 50% most non-classically responsive, $p = 0.21$, two-sided Mann–Whitney $U$ test with Bonferroni correction). However, we observed a qualitative difference in the network dynamics that

produced these errors depending on which subpopulation was inactivated. Inactivating classically responsive units led to greater response error in the readout node activity during the choice period (Supplementary Fig. 5b, d; root mean squared error during response period at 50% inactivation of most classically responsive = 0.48, 50% most non-classically responsive = 0.44, $p = 1.1 \times 10^{-6}$, $N = 24$ networks, two-sided Mann–Whitney $U$ test with Bonferroni correction). In contrast, inactivating non-classically responsive units caused a greater shift in readout node baseline activity (Supplementary Fig. 5b, e; mean shift in baseline activity at 50% inactivation of most classically responsive = $-0.25$ vs. 50% most non-classically responsive = 0.47, $p = 2.9 \times 10^{-7}$, $N = 24$ networks, two-sided Mann-Whitney $U$ test with Bonferroni correction). This suggests that while both response types are essential for proper network function, non-classically responsive units served to set boundaries which constrain overall network dynamics to the task-relevant subspace (i.e., readout node activity close to 0 except during 'go' responses) whereas classically responsive units affected dynamics within those boundaries.

We asked if these impairments resulted from silencing output connections or interfering with recurrent activity by either selectively inactivating output connections (shaped by FORCE) or recurrent connectivity (shaped by STDP). Inactivation of the output connections alone in either subpopulation impaired performance. Removing output connections from 10% of non-classically responsive output units led to a significant decrease in task performance (Fig. 3e; Supplementary Fig. 5f; removing output connections from 10% most non-classically responsive $\Delta d' = -1.2$ vs. 10% most classically responsive $\Delta d' = -0.6$, $p = 0.011$, $N = 24$ networks, two-sided Mann–Whitney $U$ test with Benjamini–Hochberg correction). For greater than 20% inactivation, silencing output connections from classically responsive units resulted in a stronger decrease in performance (Fig. 3e; Supplementary Fig. 5f; $p = 0.011$, $N = 24$ networks, two-sided Mann–Whitney $U$ test with Benjamini–Hochberg correction). This indicates that while classically responsive units contributed more to task performance via their output connections overall, a small fraction of highly non-classically responsive units were also critical for high levels of performance via their output projections. To impair the recurrent contribution of network units we replaced their recurrent activity with firing rate matched Poisson noise while leaving the activity directed towards the output node unperturbed. In contrast to silencing output connections, disabling the recurrent contributions of non-classically responsive units resulted in a larger impairment in performance than classically responsive units for all numbers of units inactivated (Fig. 3f; Supplementary Fig. 5g; disabling 10% most non-classically responsive output units $\Delta d' = -0.7$ vs. 10% most classically responsive units $\Delta d' = -0.4$, $p = 0.036$, $N = 15$ networks, two-sided Mann–Whitney $U$ test with Benjamini–Hochberg correction; disabling 50% most non-classically responsive output units $\Delta d' = -1.7$ vs. 50% most classically responsive units $\Delta d' = -1.5$, $p = 0.036$, $N = 15$ networks, two-sided Mann–Whitney $U$ test with Benjamini–Hochberg correction).

These inactivation experiments demonstrate that both classically and non-classically responsive units contribute significantly to task performance. Selective inactivation reveals that classically responsive units contribute more to task performance through their output projections while non-classically responsive units contribute primarily through their effect on recurrent activity. How do these recurrent perturbations affect the response profile distributions? Even though output weights remain fixed during inactivation, perturbing the recurrent activity of either classically or non-classically responsive units is sufficient to shift the response profiles of individual units and balance the relative output weights between classically and non-classically responsive populations to the readout node (Supplementary Fig. 5h). Unsurprisingly, recurrent inactivation of classically responsive neurons shifts the response profile distribution towards non-classical activity, as fewer classically responsive units are available

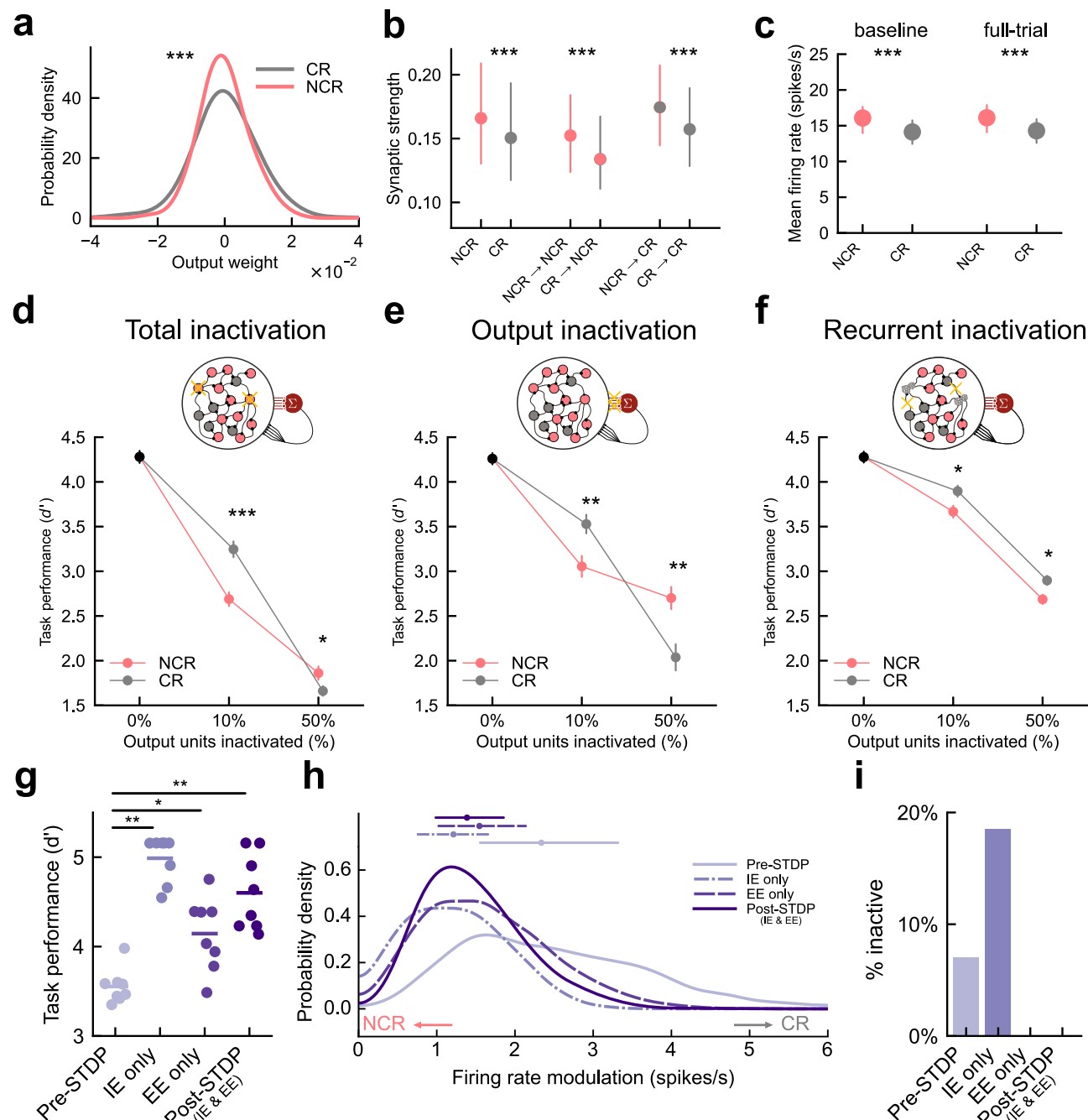

**Fig. 3 | Classically and non-classically responsive units differentially contribute to task performance and these response profiles are shaped by excitatory and inhibitory STDP. a** Output weight probability density function for non-classically (red, NCR) and classically responsive units (grey, CR). NCR vs. CR $p = 4.9 \times 10^{-22}$, $N = 8$ networks, $n = 4800$ connections, Levene's test. **b** Synaptic weights from NCRs (left, red) greater than CRs (right, grey) overall and conditioned on the target subpopulation (NCR, CR). Circles=median, lines=interquartile range. NCR vs. CR, $p < 2.0 \times 10^{-5}$; NCR → NCR vs. CR → NCR, $p < 2.0 \times 10^{-5}$; NCR → CR vs. CR → CR, $p < 2.0 \times 10^{-5}$, $N = 8$ networks, $n = 4800$ units, two-sided Hierarchical Bootstrapping test. **c** Average baseline (left) and full-trial (right) firing rates of NCRs (red) higher than CRs (grey). Circles = median, lines = interquartile range. Baseline median NCR = 16.1 vs. CR = 14.1 spikes/s, $p < 2.0 \times 10^{-5}$; full-trial NCR = 16.1 vs. CR = 14.3 spikes/s $p < 2.0 \times 10^{-5}$, $N = 8$ networks, $n = 4800$ units, two-sided Hierarchical Bootstrapping test. **d** Task performance vs. number of output units inactivated for NCRs only (red), CRs only (grey). Points = mean, bars = SEM. Inactivating 60 units i.e. 10% most NCR Δd'= −1.58 vs. CR Δd' = −0.92, $p = 1.11 \times 10^{-4}$; inactivating 300 units, i.e. 50% most NCR Δd' = −2.27 vs. CR Δd' = −2.61, $p = 0.042$, $N = 24$ networks, two-sided Mann–Whitney $U$ test with Benjamini–Hochberg correction. **e** Same as (**d**) for output weight

inactivation only (leaving recurrent connections intact). Removing output connections of 10% most NCR Δd' = −1.24 vs. CR Δd' = −0.57, $p < 10^{-5}$, $N = 24$ networks, two-sided Mann–Whitney $U$ test with Bonferroni correction. **f** Same as (**d**) for recurrent perturbation only (leaving output connections intact). Disabling recurrent connections of 10% most NCR output units Δd' = −0.74 vs. CR Δd' = −0.43, $p = 0.016$, $N = 15$ networks, two-sided Mann–Whitney $U$ test with Bonferroni correction. **g** Task performance with no STDP (pre-STDP), inhibitory-to-excitatory STDP (IE only), excitatory-to-excitatory STDP (EE only), or all STDP (post-STDP). $N = 8$ networks per group. Circles = networks, bars = means. pre-STDP vs. IE only, Δd' = 1.43 ± 0.09, $p = 0.0014$; EE only: Δd' = 0.59 ± 0.18, $p = 0.011$; Post-STDP: Δd' = 1.05 ± 0.19, $p = 0.0014$, two-sided Wilcoxon test with Bonferroni correction. **h** Output unit firing rate modulation probability density, same conditions as (**g**). Circles = median, bars = interquartile range. Pre-STDP vs. post-STDP: Δmodulation = −0.95 spikes/s, IE only: Δmodulation = −1.13 spikes/s, EE only: Δmodulation = −0.79 spikes/s, $p < 10^{-5}$ for all comparisons, $N = 8$ networks, $n = 4600$ units per condition, two-sided Mann–Whitney $U$ test Bonferroni correction. Inactive units excluded and percentages shown in (**i**). Source data are provided as a Source Data file.

to amplify the responses of other units in the network to stimulus or choice (Supplementary Fig. 5i, top). Remarkably, perturbing the recurrent activity of non-classically responsive units also shifts the response distribution away from classically responsive units indicating that the recurrent activity of non-classically responsive units collectively supports the activity of classically responsive units despite the smaller stimulus- and choice-related firing rate modulation exhibited in non-classically responsive units (Supplementary Fig. 5i, bottom). Recurrent inactivation randomizes the timing of single-unit spiking activity while maintaining the overall firing rate using a Poisson process. Because non-classically responsive units have a comparatively constant trial-averaged firing rate, this perturbation demonstrates that the spike-timing of these units enables the activity that drives the readout node in the unperturbed network.

### Effect of network parameters on ensemble diversity

Four of the main parameters in the network model are network size, fraction of inhibitory units, fraction of input units, and connection probability. We systematically varied each of those parameters (leaving other parameters fixed) to assess their effect on the distribution of classically responsive and non-classically responsive network unit response profiles in networks where STDP was active (post-STDP). Increasing network size while keeping the average number of connections per unit constant increased the proportion of classically responsive units (Supplementary Fig. 6a, left; $p < 10^{-5}$ for 2000 units vs 1000 and 500, $N = 4$ networks, $n = (1200, 2400, 4800)$ units per parameter size = (500, 1000, 2000), two-sided Mann–Whitney $U$ test with Bonferroni correction) which was primarily a result of higher variability in the choice-related responses of network units (Supplementary Fig. 6a, middle, right; $p < 10^{-5}$ all comparisons, $N = 4$ networks, $n = (1200, 2400, 4800)$ units per parameter size = (500, 1000, 2000), Levene's test with Bonferroni correction). Increasing the fraction of inhibitory units reduced the overall responsiveness (Supplementary Fig. 6b, left; $p < 10^{-5}$ all comparisons, $N = 4$ networks, $n = 2400$ units per parameter, two-sided Mann–Whitney $U$ test with Bonferroni correction) by decreasing the magnitude of both stimulus and choice-related activity (Supplementary Fig. 6b, middle, right; stimulus, $p < 10^{-5}$ all comparisons, two-sided Mann–Whitney $U$ test with Bonferroni correction; choice, $p < 10^{-5}$ all comparisons, Levene's test with Bonferroni correction). Increasing the fraction of input cells increased responsiveness (Supplementary Fig. 6c, left; $p < 10^{-5}$, $N = 4$ networks, $n = 2400$ units per parameter, all comparisons, two-sided Mann–Whitney $U$ test with Bonferroni correction.) by increasing the stimulus response of the network (Supplementary Fig. 6c, middle, right; $p < 10^{-5}$, $N = 8$ networks, $n = 4800$ units per parameter, all comparisons, two-sided Mann–Whitney $U$ test with Bonferroni correction). Finally, increasing the connection density of the network increased responsiveness (Supplementary Fig. 6d, left; $p < 10^{-5}$, $N = 8$ networks, $n = 4800$ units per parameter, all comparisons, two-sided Mann–Whitney $U$ test with Bonferroni correction) despite the fact that connection weights were scaled down as connection probability increased. This increase was driven by an increase in stimulus-related activity (Supplementary Fig. 6d, middle, right; $p < 10^{-5}$ all comparisons, $N = 8$ networks, $n = 4800$ units per parameter, two-sided Mann–Whitney $U$ test with Bonferroni correction).

### Synaptic mechanisms shape response profile distributions and task performance

Our exploration of network parameters indicated that by changing connectivity statistics—perhaps simulating the effects of synaptic plasticity mechanisms—networks can adjust the relative fraction of classically responsive and non-classically responsive units. Therefore, we next examined the sensitivity of response profiles to the details of STDP in the model. To further understand how STDP improved task performance and determine the specific role of inhibitory and excitatory STDP mechanisms, we selectively included either only excitatory-to-excitatory or inhibitory-to-excitatory plasticity mechanisms during training and evaluated the effect on task performance and response profile distributions.

Selectively enabling either only inhibitory-to-excitatory or excitatory-to-excitatory plasticity boosted task performance relative to pre-STDP networks. However, including only inhibitory-to-excitatory plasticity produced performance gains comparable to the post-STDP condition where both rules were active (Fig. 3g; $N = 8$ networks per condition, increase in task performance relative to pre-STDP with only inhibitory-to-excitatory STDP, $\Delta d' = 1.4 \pm 0.1$, $p = 0.0014$, increase in performance with only excitatory-to-excitatory STDP: $\Delta d' = 0.6 \pm 0.2$, $p = 0.011$, increase in performance with both forms of STDP: $\Delta d' = 1.1 \pm 0.2$, $p = 0.0014$, two-sided Wilcoxon test with Bonferroni correction). Plasticity in excitatory-to-excitatory and inhibitory-to-excitatory synapses were both sufficient to shift the response profile distribution towards non-classically responsive activity and create a distribution similar to that observed in the full post-STDP model (Fig. 3h; median shift in firing rate modulation relative to pre-STDP, Post-STDP: $\Delta$modulation $= -0.95$ spikes/s, IE only: $\Delta$modulation $= -1.13$ spikes/s, EE only: $\Delta$modulation $= -0.79$ spikes/s, $p < 10^{-5}$ for all comparisons, $N = 8$ networks, $n = 4600$ units per condition, two-sided Mann–Whitney $U$ test with Bonferroni correction).

Notably, including inhibitory-to-excitatory plasticity alone produced a larger number of inactive units (Fig. 3i, 'IE only'; firing rate <1 spike/trial on average). While inhibitory-to-excitatory synaptic plasticity was sufficient to improve performance to post-STDP levels, it decoupled a large fraction of network units in the process; however, in tandem with excitatory-to-excitatory plasticity performance gains can occur while all units remained engaged during task performance. Although the full network engagement maintained by excitatory plasticity was not necessary to improve performance on this task (Fig. 3g), maintenance of full network engagement effectively increases the size of the network which may have benefits in other tasks or contexts.

These plasticity mechanisms differed in how they shifted response profiles into a non-classically responsive regime. While both inhibitory-to-excitatory and excitatory-to-excitatory plasticity reduced firing rate modulation during the stimulus period (Supplementary Fig. 7a, left; median shift relative to pre-STDP, post-STDP: $\Delta$stimulus $= -0.52$ spikes/s, IE only: $\Delta$stimulus $= -0.70$ spikes/s, EE only: $\Delta$stimulus $= -0.67$ spikes/s, $p < 10^{-5}$ for all comparisons to pre-STDP, $N = 8$ networks, $n = 4600$ units per condition, two-sided Mann–Whitney $U$ test with Bonferroni correction), only excitatory-to-excitatory plasticity shifted choice modulation towards post-STDP values (Supplementary Fig. 7a, right; median shift relative to pre-STDP, post-STDP: $\Delta$choice $= -0.95$ spikes/s, $p < 10^{-5}$, IE only: $\Delta$choice $= -0.07$ spikes/s, $p = 0.078$, EE only: $\Delta$choice $= -0.97$ spikes/s, $p < 10^{-5}$, $N = 8$ networks, $n = 4600$ units per condition, two-sided Mann–Whitney $U$ test with Bonferroni correction). Furthermore, the range of both stimulus and choice-related modulation values decreased when inhibitory-to-excitatory mechanisms were included regardless of whether excitatory-to-excitatory mechanisms were present (Supplementary Fig. 7a; IE only vs. Pre-STDP and Post-STDP vs. EE only, $p < 10^{-3}$, $N = 8$ networks, $n = 4600$ units per condition, Levene's test with Bonferroni correction). This suggests that excitatory-to-excitatory plasticity shifts response profiles toward non-classically responsive activity by shifting median responses closer to zero modulation while inhibitory-to-excitatory plasticity also constrains the range of modulation observed during each trial period.

We next sought to understand how the three forms of plasticity (homosynaptic STDP, heterosynaptic balancing β, and heterosynaptic enhancement δ) adjusted the population of unitary response profiles. To do this, we selectively increased or decreased the strength of each type of plasticity relative to default values for either excitatory-to-excitatory synapses or inhibitory-to-excitatory synapses. Altering the

strength of excitatory STDP did not change the response profile distribution (Supplementary Fig. 7b, left); however, excitatory heterosynaptic mechanisms significantly modified response properties (Supplementary Fig. 7b, center, right). Strengthening excitatory heterosynaptic balancing (β) increased the firing rate modulation of network units (Supplementary Fig. 7b, center; $p < 10^{-5}$, $N = 8$ networks, $n = 4600$ units per condition, for 2x vs 0.5x strength, Kolmogorov–Smirnov test with Bonferroni correction), whereas strengthening heterosynaptic enhancement (δ) decreased modulation (Supplementary Fig. 7b, right; $p < 10^{-5}$ for 2x to 0.5x strength, $N = 8$ networks, $n = 4600$ units per condition, Kolmogorov–Smirnov test Bonferroni correction). For inhibitory plasticity, heterosynaptic enhancement (δ) increased the modulation of network units in opposition to excitatory-to-excitatory heterosynaptic enhancement (Supplementary Fig. 7c, right; $p < 10^{-5}$ for 2x vs 0.5x strength, $N = 8$ networks, $n = 4600$ units per condition, Kolmogorov–Smirnov test with Bonferroni correction). Similarly, inhibitory homosynaptic terms also increased the strength of classical responses (Supplementary Fig. 7c, left; $p < 10^{-5}$ for 2x vs 0.5x strength, $N = 8$ networks, $n = 4600$ units per condition, Kolmogorov–Smirnov test with Bonferroni correction). Strengthening inhibitory heterosynaptic balancing, β, had the effect of expanding the dynamic range of response profiles without shifting the median, resulting in a greater diversity of response types (Supplementary Fig. 7c, center; $p < 10^{-5}$ for 2x vs 0.5x strength, $N = 8$ networks, $n = 4600$ units per condition, Kolmogorov–Smirnov test with Bonferroni correction). These results suggest that while excitatory-to-excitatory homosynaptic plasticity has minimal effect on response types, each of the other synaptic mechanisms provide complementary constraints on the range and median of the response profile distribution.

## Specific local synaptic patterns predict response properties of diverse units

As STDP of excitation and inhibition shaped response profiles across the network, we next determined how individual classically and non-classically responsive units were embedded in the network by examining their synaptic input and output patterns. We analyzed the responses of the group of excitatory units which projected to the readout node since these neurons were not trivially stimulus responsive (via direct stimulus inputs) and displayed the greatest range of stimulus and choice modulation ('output units'; Supplementary Fig. 3d, center).

Although the strengths of direct (i.e., monosynaptic) inputs onto a given unit should predominantly determine the response profiles of individual units, higher order correlations may also be relevant for determining single-unit responses. To assess the contributions of direct and higher-order synaptic connections (e.g., disynaptic, trisynaptic, etc.), we adapted a recent approach[50] to systematically decompose the weight matrix into local patterns of connectivity or "motifs" (Fig. 4a), starting with small numbers of synaptic connections (lower-order motifs) and progressing to motifs with larger numbers of connections (higher-order motifs). These local patterns have been shown to predict a variety of network phenomena including cross-correlations and the dimensionality of network dynamics[51]. Motifs with larger numbers of synaptic connections are only considered present in a network if they are unlikely to occur from random combinations of motifs with fewer connections. In previous work[50,51], only network-wide averages of these synaptic patterns were required, but here we derive motifs for each individual unit which sum to produce the full network-wide motif cumulants providing a neuron-by-neuron view of synaptic structure.

There are three main classes of synaptic patterns (Fig. 4a): 'Chain motifs' represent sequential synaptic connections. 'Convergent motifs' represent two neurons that project to a common downstream output neuron separated by one or many synapses. 'Divergent motifs' represent two neurons that share an upstream input unit separated by one or many synapses. The simplest motif is a 'first-order chain motif' which simply represents the average strength of synaptic inputs or outputs of a unit. To explain the responsiveness of excitatory output units we considered synaptic patterns shared with all four subpopulations in the network: 'output units' that project connections to the readout node, 'target responsive input units' that receive stimulus current on target trials, 'non-target responsive input units' that receive stimulus current on non-target trials, and 'inhibitory units' (Fig. 4b). For example, the first-order chain motif from the inhibitory subpopulation are simply the average inhibitory synaptic inputs to a unit; the "2nd order divergent motif" between a unit and the inhibitory subpopulation would indicate that it receives synaptic inputs from the same units as inhibitory units (beyond what can be explained by chance).

Classically and non-classically responsive units demonstrated distinct patterns of connectivity to the four subpopulations of the network (output units, target responsive input units, non-target responsive input units, and inhibitory units). Examination of the monosynaptic motifs (direct synaptic inputs and outputs to the four subpopulations) revealed that non-classically responsive units received weaker inputs than classically responsive units overall, particularly from inhibitory units and target responsive units (Fig. 4c, monosynaptic inputs, change in median normalized synaptic weights for non-classically vs. classically responsive units = −0.20, $p < 10^{-5}$, $N = 24$ networks, $n = 14,400$ units total, two-sided Mann–Whitney $U$ test; Supplementary Fig. 8a, top, all comparisons of non-classically vs. classically responsive $p < 10^{-5}$, $N = 24$ networks, $n = 14,400$ units total, two-sided Mann–Whitney $U$ test with Bonferroni correction). Despite receiving weaker inputs, non-classically responsive units had stronger recurrent synaptic outputs than classically responsive units to all other excitatory subpopulations (Fig. 4c, monosynaptic outputs, change in median synaptic input weight for non-classically vs. classically responsive units = 0.13, $p < 10^{-5}$, $N = 24$ networks, $n = 14,400$ units total, two-sided Mann–Whitney $U$ test; Supplementary Fig. 8a, bottom; all comparisons of NCR vs. CR except inhibitory outputs, $p < 10^{-4}$, inhibitory outputs $p > 0.7$, two-sided Mann–Whitney $U$ test with Bonferroni correction).

Only a limited number of disynaptic input motifs differed systematically from zero, indicating that most disynaptic connections have no additional structure beyond their monosynaptic motifs (Supplementary Fig. 8b). Interestingly, the disynaptic input motifs of non-classically responsive units are closer to zero indicating that their patterns of synaptic connectivity are more local than classically responsive units—spanning only a single synaptic connection (Fig. 4c, disynaptic inputs, change in median synaptic input weight for non-classically vs. classically responsive units = −0.36; Supplementary Fig. 8b, inputs $p < 0.005$ unless labeled n.s., two-sided Mann–Whitney $U$ test with Bonferroni correction, outputs $p < 0.04$ unless labeled n.s., Levene's test with Bonferroni correction). This suggests that non-classically responsive units receive less-correlated inputs from other units in the network and therefore perform a more diverse set of computations.

Can we predict which neurons will become classically or non-classically responsive based on their local synaptic structure? To answer this, we formulated a statistical model to predict the firing rate modulation of individual units based on its individual synaptic motifs. This statistical model uses the prevalence of synaptic motifs for individual units to predict the stimulus and choice modulation of these units via a multilinear regression. These predictions are then combined to make a prediction for the overall firing rate modulation (Fig. 4d). This model successfully predicted the firing rate modulation of individual output neurons across various network types with radically different response profiles (pre-STDP, IE only, EE only, post-STDP) demonstrating that the response profile of individual units can be explained by their local

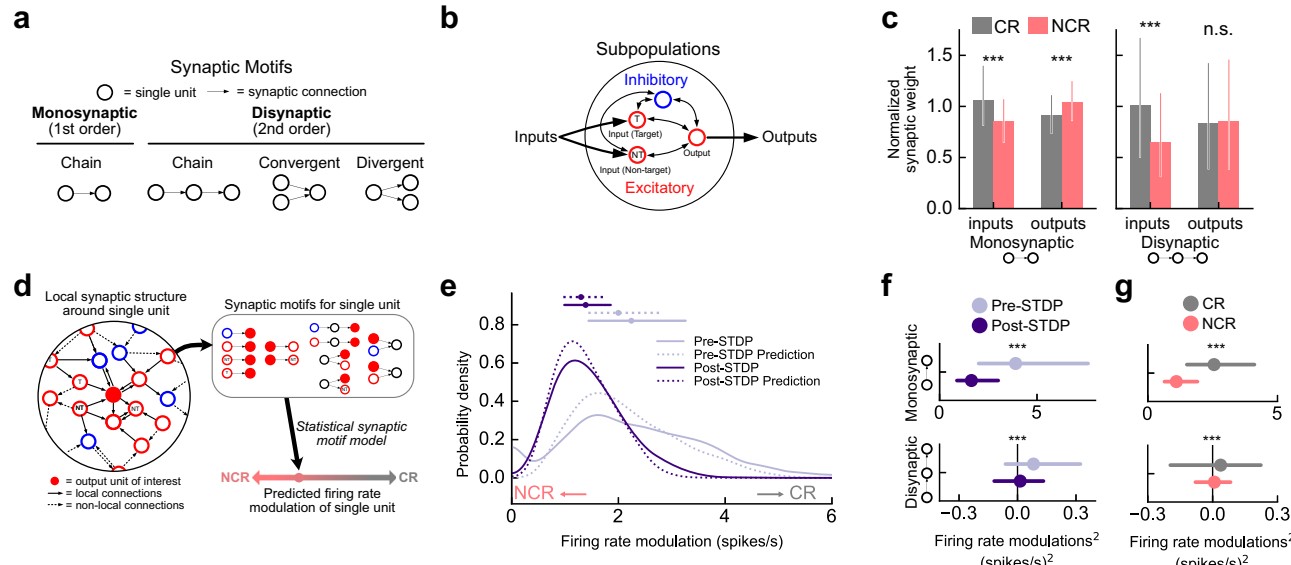

**Fig. 4 | Specific local synaptic patterns predict response properties of diverse units. a** All possible monosynaptic (first-order) and disynaptic (second-order) motifs. **b** Network schematic including 4 network subpopulations: target input units, non-target input units, output units, and inhibitory units. **c** Left, observed prevalence of all monosynaptic motifs between individual output units and all subpopulations for non-classically responsive units (NCR, red) and classically responsive units (CR, grey). Bars and lines represent medians and interquartile range, respectively. Right, same as left except for all disynaptic motifs between individual output units and all subpopulations. Change in median synaptic input weight for non-classically vs. classically responsive units, Δmonosynaptic input = −0.20, Δmonosynaptic output = 0.13, Δdisynaptic input = −0.36, $p < 10^{-5}$, Δdisynaptic output = 0.02, $p = 0.1$, $N = 24$ networks, $n = 14,400$ units total, two-sided Mann–Whitney U test. **d** Schematic of method for predicting response profile from local synaptic structure around a unit. Local structure was decomposed into a set of synaptic motifs and these motifs were used to predict the modulation of the unit. **e** Probability density of firing rate modulation for individual output units for networks without STDP (pre-STDP, solid light purple) or all STDP rules (post-STDP,

solid dark purple) along with predictions derived from statistical motif model (dotted lines). Small values of the firing rate modulation correspond to non-classical response profiles; high values correspond to classical response profiles. Summary circles and bars above distributions represent median and interquartile range, respectively. $N = 8$ networks, $n = 4800$ units per condition. **f** Contributions of monosynaptic (top) and disynaptic motifs (bottom) to single-unit firing rate modulation$^2$ for pre-STDP networks (light purple) and post-STDP networks (dark purple). Circles and bars represent median and interquartile range, respectively. Pre-STDP vs. post-STDP for monosynaptic and disynaptic, $p < 10^{-5}$, $N = 8$ networks, $n = 4800$ units per condition, two-sided Mann–Whitney U test with Bonferroni correction. **g** Contributions of monosynaptic (top) and disynaptic motifs (bottom) to single-unit firing rate modulation$^2$ for classically responsive units (CR, grey) and non-classically responsive units (NCR, red). Circles and bars represent median and interquartile range, respectively. Non-classically responsive vs. classically responsive for monosynaptic and disynaptic, $p < 10^{-5}$, $N = 8$ networks, $n = 4800$ units total, two-sided Mann–Whitney U test with Bonferroni correction. Source data are provided as a Source Data file.

patterns of synaptic weights independent of the plasticity mechanisms shaping those weights (Supplementary Fig 8c; 30-fold cross-validated stimulus mean-squared error = 0.29 ± 0.15 spikes/s, $r = 0.75 ± 0.06$, choice mean-squared error = 0.63 ± 0.43 spikes/s, $r = 0.87 ± 0.02$). Examining the output units as a whole, this statistical synaptic motif model captures the observed shift towards non-classically responsive activity post-STDP demonstrating that local synaptic plasticity can account for global changes in the response profile distribution (Fig. 4e, Supplementary Fig. 8d).

We then dissected our statistical motif model to determine the contribution of each statistically significant monosynaptic and disynaptic motif to the firing rate modulation pre- and post-STDP (Fig. 4f; Supplementary Fig. 8e, f). We found that the shift in the response profile distribution is primarily accounted for by the effect of STDP on monosynaptic motifs. This finding is consistent with our observation that "mean-matched" and "shuffled" networks where higher-order synaptic structure has been removed have response profiles very similar to post-STDP networks where this higher-order structure is present (Fig. 2g). Interestingly, STDP lessens the systematic contribution of disynaptic motifs to the firing rate modulation of individual units (Fig. 2g, bottom, Supplementary Fig. 8f) indicating that one effect of STDP is to make response characteristics "hyper-local" in that they only result from direct, monosynaptic connections.

Which synaptic motifs are relevant for determining whether a neuron is classically or non-classically responsive? We next determined the contribution of each statistically significant monosynaptic and disynaptic motif to the firing rate modulation of classically responsive and

non-classically responsive units. Monosynaptic motifs were primarily responsible for determining response profiles while disynaptic motifs played a smaller, non-systematic role (Fig. 4g, Supplementary Fig. 8g, h; all non-classically vs. classically responsive comparisons $p < 10^{-5}$, $N = 8$ networks, $n = 4800$ units total, two-sided Mann–Whitney U test with Bonferroni correction). 6 out of a possible 8 monosynaptic motifs and 6 out of a possible 16 disynaptic motifs were found to be relevant for predicting the modulation of individual units indicating that a specific set of local synaptic motifs are relevant for identifying the response properties of network units (Supplementary Fig. 8a, b, g–j; $p < 0.01$, two-sided Mann–Whitney U test with Bonferroni correction). Each of these identified motifs was relevant for the response properties of both classically responsive and non-classically responsive units. This indicated that both unit types were driven by the same types of connections. Moreover, these included connections to all subpopulations (output, target responsive input, non-target responsive input, and inhibitory) although higher-order, disynaptic connections to output and inhibitory units were of particular importance (Supplementary Fig. 8g, h; output, 3 of 4 possible disynaptic motifs, open red circles; inhibitory, 2 of 4 possible disynaptic motifs, open blue circles;). Decreases in monosynaptic inputs to non-classically responsive units and increases in their monosynaptic outputs (Fig. 4c, Supplementary Fig. 8a) all contributed to less modulated firing overall (Fig. 4g, Supplementary Fig. 8g). Moreover, the decrease in disynaptic correlations for non-classically responsive units (Fig. 4c, Supplementary Fig. 8b) resulted in a smaller disynaptic contribution to the response profiles of non-classically responsive units (Fig. 4g, Supplementary Fig. 8h).

Examining stimulus and choice-related firing rate modulation separately revealed that there were distinctions in how each sub-population affects specific task-related responses (Supplementary Fig. 8g, h). Monosynaptic connections to inhibitory and output units play a significant role in both stimulus and choice-related responses, however connections to input units only contribute significantly to stimulus responses. Furthermore, disynaptic connections play a relatively larger role in choice-related responses than stimulus responses indicating that higher-order structure may be more important in transforming stimulus-related into choice-related activity. Thus, local synaptic structure predicted the response profile of individual units with direct monosynaptic connections having the largest effect. The response properties of classically and non-classically responsive units were driven by differences in how they connect to all network sub-populations rather than one in isolation (Fig. 4c, g, Supplementary Fig. 8a, b, g, h). Specifically, non-classically responsive units resulted from weakened input from all subpopulations and smaller disynaptic correlations rather than increased inhibition or weakened inputs from input units. Interestingly, non-classically responsive units had stronger recurrent connections to the rest of the network than classically responsive units. This difference is consistent with the greater impact on network performance resulting from inactivating recurrent connections of non-classically responsive units (Fig. 3f).

## Predicting single neuron response profiles recorded in vivo

Our model makes several predictions about the relationship between synaptic inputs and output spiking responses. Specifically, we hypothesized that neuronal response type (i.e., classically responsive or non-classically responsive) can be determined from average synaptic input strengths (Fig. 4c–f). We made in vivo whole-cell voltage-clamp recordings from neurons of the auditory cortex of mice during the go/no-go frequency recognition task (Fig. 5a; $n = 12$ neurons from $N = 5$ mice). We measured excitatory and inhibitory synaptic currents (E/IPSCs) during behavior, and for some cells ($n = 4$ neurons from $N = 3$ mice), we were able to record both the spiking activity in cell-attached mode prior to breaking into the cell to record postsynaptic currents (Fig. 5b, c, Supplementary Fig. 9a). For those cells where only synaptic currents were recorded, we used a straightforward integrate-and-fire model to simulate their spiking activity based on the experimentally measured currents over individual trials, including during inter-trial intervals (Supplementary Fig. 9b). The firing rate modulation of each neuron (and thus degree to which each neuron was classically responsive or non-classically responsive) was then calculated by comparing baseline spiking activity to activity during stimulus presentation. The parameters chosen for the integrate-and-fire simulation were based on those neurons where both synaptic input and spiking output data were available for the same neuron (Fig. 5b, c, right). This simulation accurately captured the firing rate modulation of cells where spiking data was available (Fig. 5d; Pearson's $r = 0.85$) and reproduced a distribution of responses similar to those directly measured from cell-attached recordings (Fig. 5e; $p = 0.41$, two-sided Kolmogorov–Smirnov test). These results confirm that our simulations based on recorded input currents accurately reproduce the modulation of neurons where spiking outputs were not available.

Our spiking RNN also predicts that non-classically responsive units have weaker average inputs overall. We also observed this in the whole-cell recordings in vivo; non-classically responsive neurons had significantly weaker inhibitory and excitatory inputs than classically

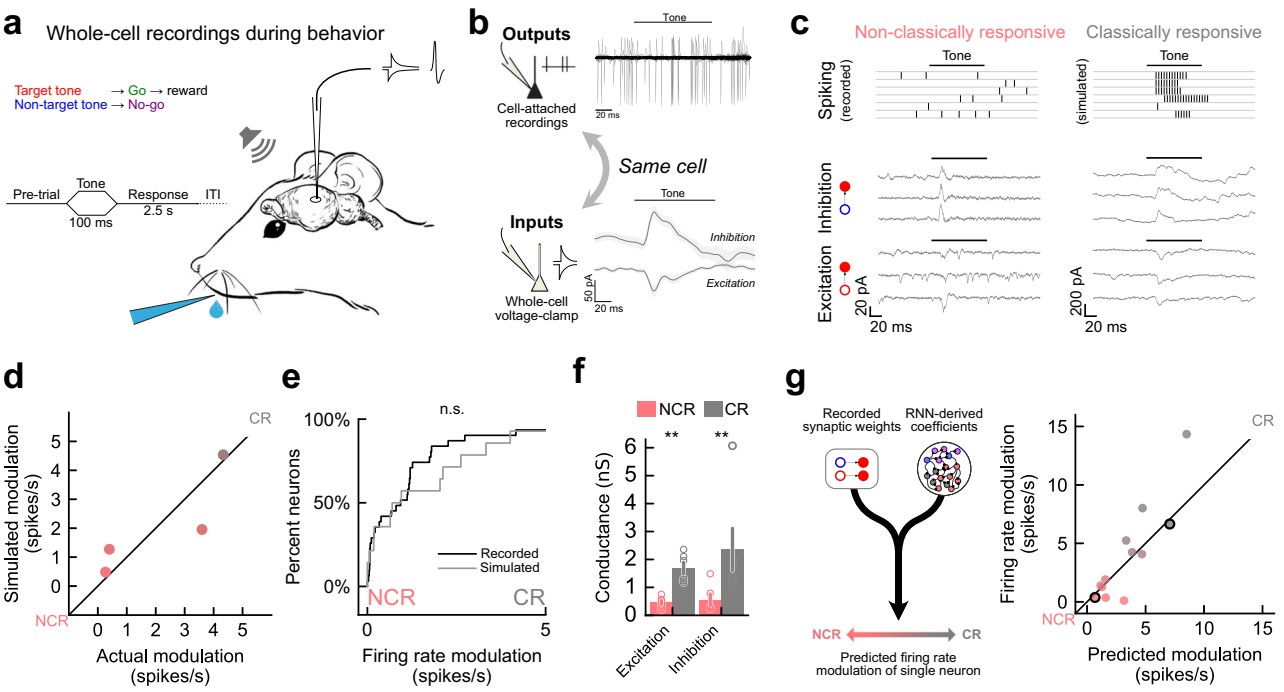

**Fig. 5 | RNN-derived statistical motif model predicts in vivo response profiles.**
**a** Schematic of behavioral task and whole-cell recording set-up during behavior.
**b** An example cell where cell-attached recording was used to first measure spiking outputs before breaking into the cell to record synaptic currents (E/IPSCs).
**c** Example recordings from a non-classically responsive neuron (left) and a classically responsive neuron (right). For non-classically responsive unit recorded spike times are shown; for classically responsive unit simulations are shown. **d** Simulated firing rate modulation versus actual modulation for 4 neurons where spiking outputs and synaptic currents were both recorded (Pearson's $r = 0.85$, $n = 4$ neurons from $N = 3$ mice). **e** Comparison of cumulative distribution function from cell-

attached recordings (black) and leaky-integrate-and-fire simulation (grey). No significant difference was observed ($p = 0.41$, two-sided Kolmogorov–Smirnov test).
**f** Comparison of average synaptic currents for non-classically responsive neurons (red) versus classically responsive neurons (black). Bars indicate means and SEM (NCR vs CR, ΔExc = −72%, $p = 0.002$, ΔInh = −76%, $p = 0.007$, $n = 12$ neurons from $N = 5$ mice, two-sided Mann–Whitney $U$ test). **g** Recorded/simulated modulation was compared to predictions based on coefficients from the RNN-derived motif statistical model, neurons from (**c**) circled in black (mean-squared error = 2.2 spikes/s, Pearson's $r = 0.94$, $n = 12$ neurons from $N = 5$ mice). Source data are provided as a Source Data file. Artwork in (**a**) by Shari E. Ross.

responsive cells (Fig. 5f, NCR vs CR, ΔExc = −72%, $p = 0.002$, ΔInh = −76%, $p = 0.007$, $n = 12$ neurons from $N = 5$ mice, two-sided Mann-Whitney U test). To evaluate whether the relationship between average synaptic inputs and spiking outputs observed in our model holds in vivo, we used parameters derived exclusively from our RNN to predict the firing rate modulation of individual neurons in vivo. These parameters were coefficients representing the relative contribution of average inhibitory and excitatory conductance to firing rate modulation during behavior and were combined with measured average synaptic conductances from cells in vivo to predict their modulation (Fig. 5g, left). The parameters were obtained from a simplified version of the same statistical model previously used to predict modulation of units in the RNN using their synaptic motifs (Fig. 4; see Methods). Comparing the modulation inferred directly from the trial-by-trial in vivo data with the one predicted by our model, we found the statistical model derived from the RNN predicts the firing modulation of individual neurons in vivo to a high degree of accuracy for neurons over a range of classically responsive and non-classically responsive firing rate modulations (Fig. 5g, right; mean-squared error = 2.2 spikes/s, Pearson's $r = 0.94$, $n = 12$ neurons from $N = 5$ mice).

To test whether the RNN-derived predictions captured detailed aspects of the trial-by-trial dynamics, we compared these predictions to modified simulations which kept average conductance values fixed while rescaling trial-by-trial conductance dynamics relative to this baseline (peak conductance values were rescaled to be closer to the mean value by a factor of 2). This ensured that the RNN-based predictions would remain unchanged while simultaneously altering the trial-by-trial currents used for simulation. The modulations produced by these rescaled dynamics were systematically lower than the RNN-derived predictions indicating that the RNN-derived coefficients capture non-trivial features of the original trial-by-trial dynamics in vivo (Supplementary Fig. 9c, mean-squared error = 3.1 spikes/s, Pearson's $r = 0.85$). Next, we checked whether the relationship between synaptic inputs and output modulation derived from our model was statistically meaningful by randomly shuffling the modulation values for RNN units and deriving coefficients that attempt to predict these random values from the synaptic structure. Using these shuffled RNN coefficients on the experimental data significantly reduced the accuracy of the predicted modulations, demonstrating that the success of our RNN-derived predictions was not due to chance (Supplementary Fig. 9d, mean-squared error = 4.9 spikes/s, Pearson's $r = 0.3$).

Finally, to assess whether the inclusion of STDP in our model was necessary to predict the relationship between synaptic input and firing rate modulation, we derived parameters using only networks without STDP (Pre-STDP). If the input/output relationship observed did not depend on the synaptic structure induced by STDP, we would expect these networks to be equally predictive for neurons in vivo. While the predictions using Pre-STDP networks alone were correlated with the observed firing rate modulation they were systematically lower than the true value implying that STDP is required to correctly recapitulate the relationship between inputs and outputs observed in vivo (Supplementary Fig. 9e, mean-squared error = 3.1 spikes/s, Pearson's $r = 0.87$). These findings indicate that our full RNN model including STDP successfully recapitulates the connection between synaptic structure and spiking response properties observed in vivo.

## Discussion

Our experimentally motivated spiking RNN model revealed that the diversity of neuronal response profiles is constrained by local synaptic structures and shaped by synaptic plasticity mechanisms. This model sits at the nexus between two recent trends in neural network modeling: First, recent work has successfully extended general-purpose learning algorithms (e.g., FORCE) designed for rate-based networks to networks with spiking units[37,52–54]. Second, there has been renewed interest in using RNNs to understand the neuronal ensembles important for learning, memory storage, and task performance[55,56] and the role of biologically motivated synaptic plasticity rules in forming these ensembles[34,35,57]. Here, we have combined these approaches in a novel spiking RNN to investigate how they shape the synaptic organization underlying heterogeneous neuronal responses that drive task performance. This produced a 'synaptic signature' in terms of the monosynaptic/disynaptic connections to any given neuron, related to how that cell was wired into the network, which learning rules helped shape those connections, and the function of that cell for task performance.

Our model successfully captures several observations regarding sensory encoding in the auditory cortex. First, it accurately captures the distribution of response profiles observed in vivo. It qualitatively reproduces a wide range of response profiles spanning non-classically and classically responsive units and specifically captures the in vivo rat modulation distribution in detail (Fig. 2e). Moreover, we have demonstrated that STDP parameters are capable of altering both the median and range of the resulting neural response distribution suggesting that differences in STDP parameters may be responsible for differences in the experimentally observed response distributions between rats and mice (Supplementary Fig. 7b, c). Second, both classically and non-classically responsive units encode task information in their ISI distribution as has been observed experimentally[16]. Critically, our model predicts the relationship between the synaptic input to a single unit embedded in a network performing a task and its observed task-related response properties. Moreover, it makes the key prediction that both classically and non-classically responsive units in the auditory cortex can contribute to task performance such that perturbation of either unit type should lead to classification impairments. Perturbation experiments in rodents have revealed that the contribution of the auditory cortex to perceptual decision making depends on details of the task design and difficulty[38]. Muscimol inactivation studies suggest that auditory cortex is required for auditory stimulus classification tasks such as the one considered here[7,16,39,58]. However, lesion and optogenetic inactivation studies show that the auditory cortex is not required for similar tasks[38,59,60]. The difference between these types of loss of function experiments may still require appropriate controls for unambiguous interpretation[38]. Our model incorporates essential biological features (spike timing, synaptic dynamics, and task performance) that, going forward, should enable us to quantify the cortical contributions (or lack thereof) to specific behaviors and tasks.

Notably, there are other important dimensions that might account for the biological basis of diverse response profiles such as dendritic geometry. It could be that non-classically responsive neurons have inputs organized differently than classically responsive neurons resulting in a differential somatic response via cable filtering[61]. Intrinsic neuron properties may also change the dynamic behavior of single neurons. For example, integrator (class I) versus resonator (class II) neurons may respond to similar synaptic inputs with distinct spike patterns[62,63].

Previous work focused on rate-based networks where the network architecture is modified through backpropagation-through-time rather than biologically motivated plasticity rules. Instead, here we analyze the contribution of neural units and populations within a spiking network with biologically motivated plasticity rules such as STDP. We believe that our approach will help advance the new wave of 'NeuroAI', using RNN models to understand how the population dynamics of a single network can successfully perform multiple tasks[55,56].

Our model allows us to probe aspects of network function that would be challenging to address experimentally. In particular, perturbation of non-classically responsive units reveals that disrupting the spike times of units with limited firing rate modulation can nevertheless have a significant impact on task performance. (Fig. 3, Supplementary Fig. 5). Although inactivation does not reveal a categorical

difference between the effect of perturbing non-classically and classically responsive units, it is remarkable that disrupting the timing of recurrent activity from non-classically responsive units has a greater impact on network performance than disruption of classically responsive activity (Fig. 3f). This finding highlights the importance of temporal coding (in addition to rate coding) for interpreting and explaining the network dynamics which drive behavior. Even small changes in network dynamics—e.g. adding, subtracting, or rearranging one or a few spikes via short-term plasticity or STDP—can have significant ramifications for the network activity driving behavior[64–68].

In principle and in practice, any neuron—or even every neuron—can contribute in some way to overall network dynamics and behavioral performance, regardless as to the degree of overt firing rate changes (i.e., classical responsiveness). Both classically and non-classically responsive units directly affected network task performance, and diverse neuronal responses throughout a population are likely essential for robustly supporting the dynamics required for sensorimotor transformations and adaptive behaviors. This diversity might also enable evidence accumulation, decision making, working memory[15], and learning in spiking RNNs[69], enabling neural processing and flexible computations over a range of spatiotemporal scales throughout the brain.

## Methods

### Electrophysiological recordings during behavior

All animal procedures were performed in accordance with National Institutes of Health standards and were conducted under a protocol approved by the New York University School of Medicine Institutional Animal Care and Use Committee. Animals were housed at room temperature (~22 °C) with a relative humidity of ~45% and a light-dark cycle of 12 h on/off (6:30 AM–6:30 PM light). All recordings were made in a dark, single-walled sound attenuating chamber that provided sound isolation and allowed for precise control over the acoustic environment. For single-unit recordings in freely moving rats, 5 adult male and 6 adult female Sprague-Dawley rats were trained to perform a go/no-go frequency recognition task as previously described[16,58,70]. Animals were trained to respond via nosepoke to a target tone (4 kHz) for food reward and to withhold their response to nontarget tones (0.5, 1, 2, 8, 16, 32 kHz). The response period was 2.5 sec and false alarms resulted in a 7 s timeout. All pure tones played were 100 ms duration, 3 ms cosine on/off ramps, at 70 dB sound pressure level (SPL). Behavioral events (stimulus delivery, food delivery, and nosepoke detection) were monitored and controlled with a custom-programmed microcontroller (Med Associates). All behavioral events were registered as TTL pulses. Speaker outputs were also recorded separately to ensure stimulus identity and timing alignment with outputs recorded via TTL. Nose pokes were detected by an infrared beam break. These behavioral events were then integrated with the neural recording signal using a signal processor (Blackrock Microsystems, Cerebrus Neural Signal Processor, 30 kS/s sampling rate). After animals reached behavioral criteria (percent correct: ≥70%, $d'$: ≥1.5), rats were anesthetized with ketamine (40 mg/kg) and dexmedetomidine (0.125 mg/kg) and implanted with tetrode microdrive arrays (Versadrive-8 Neuralynx) in right auditory cortex as previously described[16]. Prior to implantation with a multielectrode array, primary auditory cortex (A1) was coarsely mapped by recording multiunit responses using a tungsten electrode. Tetrodes were advanced around 60 μm the day prior to recordings to a maximum of 2 mm from the pial surface and recordings were made from middle to deep cortical layers (putative cortical layers 3 to 6). Signals were filtered between 250 Hz and 5 kHz and digitized at 30 kHz. All above-threshold events with signal-to-noise ratios > 3:1 were stored for offline spike sorting. Single units were identified on each tetrode using OfflineSorter (Plexon Inc) by manually classifying spikes projected as points in 2D or 3D feature space. Clustering quality was evaluated using the Isolation Distance and $L_{ratio}$ sorting quality metrics. The animals and units shown were previously described[16].

For cell-attached recordings in head-fixed mice, 4 adult male and 6 adult female C57Bl/6 mice were anesthetized with isoflurane (3% during induction, 2% during surgery), and a custom-designed stainless steel headpost was affixed to the skull with dental cement (Metabond). Following 7+ days of recovery and 7+ days of subsequent water restriction, mice were trained on a go/no-go frequency-recognition task[7]. Animals were trained to respond to the target tone (11.2 kHz) by licking for water reward and to withhold responses to the non-target tone (5.6 kHz). Acoustic stimuli were 100 ms duration, 3 ms cosine on/off ramps, at 70 dB SPL. Animals had 2.5 s to respond and false alarms resulted in a 7 s timeout. Behavioral events (stimulus delivery, water delivery, and lick detection) were monitored and controlled by custom-written programs in MATLAB that interfaced with an RZ6 processor (Tucker-Davis Technologies). All behavioral events were registered as TTL pulses. Speaker outputs were also recorded separately to ensure stimulus identity and timing alignment with outputs recorded via TTL. After animals reached behavioral criteria (percent correct: ≥70%, and $d'$: ≥1.5), mice were anesthetized with isoflurane and a 3 mm diameter glass cranial window with a small 200 μm hole to allow for pipette access was implanted over auditory cortex (1.75 mm anterior to the lambdoid suture). After recovery, in vivo cell-attached or whole-cell recordings were obtained from neurons located 300–900 μm below the pial surface during behavior as previously described[7]. We ensured that the recordings were made from primary auditory cortex (A1) based on the tonotopic gradient and short-latency tone onset responses. Recordings were made in a sound-attenuation chamber (Eckel) using a Multiclamp 700B amplifier (Molecular Devices). For voltage-clamp experiments, whole-cell pipettes (5–7 MΩ) contained (in mM): 130 Cs-methanesulfonate, 4 TEA-Cl, 4 MgATP, 10 phosphocreatine, 10 HEPES, 1 QX-314, 0.5 EGTA, 2 CsCl, pH 7.2, $R_i = 257 \pm 92$ MΩ (s.d.), $R_s = 67 \pm 59$ MΩ (s.d.). Data were filtered at 2 kHz, digitized at 20 kHz, and analyzed with Clampfit 10 (Molecular Devices). For classically responsive neurons, $R_i = 278 \pm 112$ MΩ (s.d.), $R_s = 69 \pm 76$ MΩ (s.d.) and for non-classically responsive neurons, $R_i = 234 \pm 84$ MΩ (s.d.), $R_s = 64 \pm 54$ MΩ (s.d.). Cells were held at −70 mV to measure EPSCs and above 0 mV for IPSCs.

### Characterization of response profiles using a continuous measure

To comprehensively characterize spiking responses, we used a continuous measure of responsiveness which generalizes the binary classification (classically vs. non-classically responsive) we used previously[16]. Our continuous measure of responsiveness quantified the degree to which a cell exhibited firing rate changes during both the stimulus and choice periods. For the experimental data, we calculated the trial-averaged change in firing rate for each neuron during stimulus presentation relative to intertrial baseline, $R_{st}$, using a 150 ms window from stimulus onset to 50 ms post-stimulus to capture offset responses. The trial-averaged change in firing rate prior to behavioral choice, $R_{ch}$, was calculated using a window spanning the 500 ms prior to behavioral response on 'go' trials and 500 ms prior to the average behavioral response on 'no-go' trials. For spiking RNN units, stimulus modulation was calculated using the 100 ms stimulus period and choice modulation was calculated using the 100 ms choice period.

Our overall measure of firing rate modulation, $R$, combined these two terms in quadrature so that both stimulus- and choice-related firing rate changes must be small for a unit to be characterized as having a low firing rate modulation (i.e., have a more non-classically responsive response profile),

$$R^2 = R_{st}^2 + R_{ch}^2. \tag{2}$$

This measure captures the detailed firing rate modulation of individual units during both the stimulus and response periods. It provides information about the degree to which a unit is classically responsive such that values close to 0 are only possible when a unit is non-classically responsive (e.g., a unit with firing rate modulation of 0.1 spikes/s would be classified as a non-classically responsive unit whereas we would consider 5 spikes/s as highly classically responsive).

## Discrete characterization of classically and non-classically responsive units

Statistical identification of classically and non-classically responsive units followed our previous methods[16]. We used two positive statistical tests for non-classical responses to establish lack of responses during either the stimulus and/or response periods. The test compared the number of spikes during each of these windows to inter-trial baseline. Given that spike counts are discrete, bounded, and non-normal, we used subsampled bootstrapping to evaluate whether the mean change in spikes during tone presentation or the response period was sufficiently close to zero (in our case, 0.2 spikes). We subsampled 90% of the spike count changes from baseline, calculated the mean of these values, and repeated this process 5000 times to construct a distribution of means. If 95% of the subsampled mean values were between −0.2 and 0.2, we considered the cell sensory non-classically responsive ($p < 0.05$). This is a conservative, rigorous method for identifying a cell or unit as being 'non-classically responsive'[16].

## Spiking recurrent neural network model

To study the origin and functional contributions of diverse neural response profiles, we simulated a spiking neural network of 1,000 sparsely connected (5% connection probability) leaky integrate-and-fire units (800 excitatory, 200 inhibitory) with current-based synaptic input. All parameters listed in Supplementary Table 1. The temporal evolution of the membrane voltage $V_i$ of unit $i$ is

$$\tau_m \frac{dV_i}{dt} = V_i - V_r + I_{Ei}(t) - I_{Ii}(t) + I_0 \tag{3}$$

where $\tau_m = 20$ ms is the membrane time constant, $V_r = -65$ mV is the resting membrane potential, $I_{Ei}(t)$ and $I_{Ei}(t)$ are the recurrent EPSCs and IPSCs to unit $i$, respectively, and $I_0$ is the leak current. Upon reaching a threshold value of $V_{th} = -55$ mV the unit emits an action potential and its membrane voltage is reset to $V_r$.

EPSCs and IPSCs decay exponentially with time constants of $\tau_E = 20$ ms and $\tau_I = 20$ ms respectively such that if neuron $j$ synapses onto neuron $i$ with synaptic weights $W_{ij}$ and fires action potentials at times $\{t_{jk}\}$ the excitatory and inhibitory currents are

$$I_{Ei}(t) = \sum_{k,j \in E} W_{ij}(t_{jk}) \Theta(t - t_{jk}) e^{-(t - t_{jk})/\tau_E}$$
$$I_{Ii}(t) = \sum_{k,j \in I} W_{ij}(t_{jk}) \Theta(t - t_{jk}) e^{-(t - t_{jk})/\tau_I} \tag{4}$$

Where $\Theta$ is the Heaviside step function. Non-zero values of the weight matrix $W_{ij}$ were initialized to a uniform distribution between 0 and a maximum value. The initial mean value was set differently for each connection type in our network (excitatory-to-excitatory, inhibitory-to-excitatory, excitatory-to-inhibitory, inhibitory-to-inhibitory) to ensure initial chaotic dynamics amenable to FORCE training (see below 'Task and FORCE training'). Initial output weights scaled by the square root of the average number of incoming connections from each subpopulation (excitatory or inhibitory):

$$W_{E/I} = \frac{W_{0E/I}}{\sqrt{p_{con}N_{E/I}}}. \tag{5}$$

Excitatory-to-inhibitory and inhibitory-to-inhibitory weights remained fixed throughout the simulation. In contrast, excitatory-to-excitatory and inhibitory-to-excitatory weights were modified by a form of spike timing plasticity (see below 'Synaptic plasticity mechanisms').

## Task and FORCE training

The network was trained on a go/no-go stimulus classification task similar to that used experimentally for rats and mice[16,58,70]. During a 100 ms stimulus period, the network was stimulated with one of seven possible inputs (corresponding to frequencies of 0.5, 1, 2, 4, 8, 16, 32 kHz), and trained to produce an output during the subsequent 100 ms response period if the input was the 4 kHz target tone, while remaining at baseline for all other non-target tones. In between each 200 ms trial were inter-trial intervals randomly chosen from a uniform distribution between 100 and 400 ms.

A subset of excitatory units were designated as input units ($N_{in} = 200$) that received additional current during the stimulus period. Our stimuli were represented using a place code. For each stimulus $s$ a fixed subset of units $G_s$ received an additional fixed current $I_{in}$ while all other input units received no additional current:

$$I_{inj} = \begin{cases} I_{in} & \text{if } j \in G_s \\ 0 & \text{if } j \notin G_s \end{cases} \tag{6}$$

The input current magnitude, $I_{in}$, was minimized while ensuring that the network could still perform the task. The remaining excitatory units were output units ($N_{out} = 600$) which projected an additional set of weights, $w$, to a readout node whose activity, $z_{out}(t)$, represents the response of the network. The activity of the readout node is a weighted sum of the activity of all output units smoothed by an exponential kernel with time constant $\tau_{out} = 100$ ms. If output unit $i$ fires action potentials at times $\{t_{ik}\}$, its smoothed output is

$$s_i(t) = \sum_k \frac{1}{\tau_{out}} \Theta(t - t_{ik}) e^{-(t - t_{ji})/\tau_{out}} \tag{7}$$

and the readout node activity is calculated as:

$$z_{out}(t) = \sum_{i \in E_{out}} w_i(t) s_i(t). \tag{8}$$

The network was trained to produce a burst in readout node activity when the target stimulus was presented ($s = T$) and remain at baseline for all other stimuli ($s \neq T$) using a version of FORCE training adapted to spiking recurrent neural networks[36,37]. Specifically, $z_{out}(t)$ was trained to approximate:

$$f_{out}(t) = \begin{cases} \Theta(t - 100) \sin(\frac{(t-100)\pi}{100}) & s = T \\ 0 & s \neq T \end{cases}. \tag{9}$$

FORCE training requires that network output units receive feedback from the readout node in the form of an additional current. If $I_{FB\,i}(t)$ is the feedback current to readout node $i$ then

$$I_{FBi}(t) = Q\eta_i z_{out}(t) \tag{10}$$

where $Q$ is the network-wide feedback strength, $\eta_i$ is the fixed feedback weight onto output unit $i$ chosen from a uniform distribution between −1 and 1.

During FORCE training, the network iteratively modified the output weights $w_i$ to reduce the error between the network output and the desired output function: $e(t) = f_{out}(t) - z_{out}(t)$. Updates to the output weights were made at random times with an average interval of $T_{FORCE} = 4$ ms. Using bold font to denote vectors and matrices, the

learning rule for a given updated interval of $\Delta t$ is:

$$\boldsymbol{w}(t) = \boldsymbol{w}(t-\Delta t) - \frac{e(t)\boldsymbol{P}(t)\boldsymbol{s}(t)}{1 - \boldsymbol{s}^T(t)\boldsymbol{P}(t)\boldsymbol{s}(t)} \qquad (11)$$

where $P(t)$ is the network estimate for the inverse of the correlation matrix and is updated according to

$$\boldsymbol{P}(t) = \boldsymbol{P}(t-\Delta t) - \frac{\boldsymbol{P}(t-\Delta t)\boldsymbol{s}(t)\boldsymbol{s}(t)^T\boldsymbol{P}(t-\Delta t)}{1 + \boldsymbol{s}(t)^T\boldsymbol{P}(t-\Delta t)\boldsymbol{s}(t)}. \qquad (12)$$

Initial weights, $w(0)$, were chosen from normally distributed gaussian values with a standard deviation of $k_O = 0.1 / N_{out}$ and $P(0) = I$ where $I$ is the identity matrix and $\lambda$ is a model parameter which acts as a regularizer such that:

$$\boldsymbol{P}(t)^{-1} = \int_0^t \boldsymbol{s}(t)\boldsymbol{s}(t)^T \, \mathrm{d}t + \lambda \boldsymbol{I}. \qquad (13)$$

In our network, $\lambda = 1$. Once training was complete output weight $w$ remained fixed.

After output weights stabilized and networks reached asymptotic performance (i.e., after 2000 to 5000 trials), all weights were fixed and additional trials were run to evaluate network performance. Network performance was calculated by examining the integrated output activity during the response period (100 to 200 ms):

$$R = \int_{100}^{200} z_{\mathrm{out}}(t) \, \mathrm{d}t \qquad (14)$$

An optimal threshold was calculated to distinguish the integrated output on target versus non-target trials by regressing the trial type (target vs. non-target) against the integrated output activity, $R$, using logistic regression. Integrated outputs above/below this threshold were considered go/no-go responses. All d' were calculated using these response designations.

## Synaptic plasticity mechanisms

FORCE training modified the output weight vector $w$ while the recurrent weight matrix was modified by homo- and heterosynaptic forms of STDP. All recurrent inputs to excitatory cells were plastic and we used distinct homosynaptic mechanisms for excitatory-to-excitatory synapses and inhibitory-to-excitatory synapses. Inputs to inhibitory units remained fixed throughout training and testing.

**Homosynaptic excitatory-to-excitatory plasticity.** In our model, homosynaptic excitatory-to-excitatory plasticity follows a standard Hebbian model[40,71] in that when a presynaptic cell fired before a post synaptic cell, long-term potentiation (LTP) occurred and when a postsynaptic cell fired before a presynaptic cell, long-term depression (LTD) occurred. The rule can be expressed as

$$\begin{aligned} \Delta W_{ij}(t) &= A W_{ij}(t) z_{+j}(t) & \text{when postsynaptic cell } i \text{ fires} \\ \Delta W_{ij}(t) &= -B W_{ij}(t) z_{-i}(t) & \text{when presynaptic cell } j \text{ fires} \end{aligned} \qquad (15)$$

Where $z_{+j}(t)$ and $z_{-j}(t)$ are each increased by 1 each time neuron $j$ fires an action potential and exponentially decay with a time constant of $\tau_+ = \tau_- = 20$ ms and $A = 0.001$ and $B = 0.00105$ represent the strength of LTP and LTD respectively.

**Homosynaptic inhibitory-to-excitatory plasticity.** Inhibitory-to-excitatory homosynaptic plasticity was based on experimental[42] and theoretical[43] work demonstrating that synapses between inhibitory and excitatory auditory cortical cells undergo LTP when spike timing is synchronous regardless of order and undergo LTD when firing is

asynchronous,

$$\begin{aligned} \Delta W_{ij}(t) &= \eta_I \, W_{ij}(t) \, (z_{Ij}(t) - \alpha) & \text{when postsynaptic cell } i \text{ fires} \\ \Delta W_{ij}(t) &= \eta_I \, W_{ij}(t) \, z_{Ii}(t) & \text{when presynaptic cell } j \text{ fires} \end{aligned} \qquad (16)$$

where $\eta_I = 0.001$ is the overall strength of inhibitory-to-excitatory plasticity, $\alpha$ represents the strength of LTD (see "Training Protocol" for more information). $z_{I\,j}(t)$ increases by 1 each time neuron $j$ fires an action potential and exponentially decays with a time constant of $\tau_I = 5$ ms. This time constant was set to match the crossover time scale of LTP to LTD observed experimentally[42].

**Heterosynaptic inhibitory-to-excitatory and excitatory-to-excitatory plasticity.** Given the instability of pairwise Hebbian excitatory plasticity[34,72], we included two forms of heterosynaptic plasticity on excitatory-to-excitatory and inhibitory-to-excitatory synapses based on previous spiking RNN studies[35] which would (1) systematically weaken all presynaptic weights to prevent any one presynaptic connection from dominating (heterosynaptic balancing)

$$\Delta W_{ij}(t) = -\beta W_{ij}(t) (z_{-j}(t))^3 \quad \text{when postsynaptic cell } i \text{ fires} \qquad (17)$$

and (2) systematically strengthen postsynaptic weights to prevent a weakened synapse from dropping out entirely (heterosynaptic enhancement)

$$\Delta W_{ij}(t) = \delta \quad \text{when presynaptic cell } j \text{ fires}. \qquad (18)$$

## Training protocol

In general, networks were trained until output weights, recurrent weights, and network behavior stabilized (2000–5000 trials). In addition to the dynamic mechanisms listed above the network-wide bias current, $I_O$, was tuned so that inhibitory firing was on average $r_I \approx 20$ spikes/s across the population. This was accomplished via a learning rule which adjusted the bias current after each trial by

$$\Delta I_0 = \eta_r (R_I - r_I) \qquad (19)$$

where $\eta_r = 0.005$ is the bias current learning rate and $R_I$ is the average firing rate of all inhibitory cells.

Given these three plasticity mechanisms (STDP, FORCE, and bias current adjustment) training proceeded in a staged manner to ensure learning for each mechanism occurred in a regime where major initial changes induced by other mechanisms had subsided and subsequent plasticity could be regarded as quasi-stable. (1) Initially only bias current adjustment was active in order to set inhibitory rates to their desired firing rate prior to activating activity dependent plasticity mechanisms (STDP). (2) After 40 trials, all STDP mechanisms were activated, and bias currents were permitted to keep adjusting to maintain inhibitory rates near their target value. During this period, inhibitory plasticity adjusted excitatory firing rates towards a target value set, $r_E = 10$ spikes/s. This value is determined via the strength of inhibitory-to-excitatory LTD (Eq. 15) by the equation[43]

$$r_E = \frac{\alpha}{2\tau_I}. \qquad (20)$$

(3) After an additional 60 trials (trial 100), FORCE training was activated once inhibitory and excitatory rates had reached their target values. STDP and bias current adjustment was maintained. (4) After an additional 100 trials (trial 200), bias current adjustment ceased and STDP and FORCE training remained active simultaneously for the remainder of the training session. The observed boost in performance occurred when STDP and FORCE were simultaneously active for at

least ~1000 trials (Supplementary Fig. 2d) indicating that the two mechanisms can interact productively.

## Simulation details and code

All simulations were carried out via an event-based simulator which integrated all dynamical variables between spiking events. All simulations were all conducted in Julia 1.7.2 with standard packages and custom written code to run simulations[73] (see GitHub repository at https://github.com/albannalab/InsanallyAlbanna-NatComm-2024/). See Brette et al. for a description of the method and comparison with other methods[74] and Engelken for implementation details[75]. Effectively, the main loop of the simulator runs as follows (all quantities beginning with a capital letter below represent arrays calculated across the network:

```
WHILE t_current <t_trial:
    # Determining input currents
    I_in = Input current
    I_force = FORCE feedback current
    I_new = I_background + I_old + I_in + I_force + PSPs
    # Integrating to next spike time
    Time_to_next_spike  =  INTEGRATE_MEMBRANE_EQUA
    TIONS(I_new, I_old)
    delta_t = min(Time_to_next_spike)
    spiking_idx = argmin(Time_to_next_spike)
    t_current += delta_t
    # Updating PSPs, output, recurrent weights, and out
    put weights
    Network_activity[spiking_idx] += t_current
    PSPs = UPDATE_PSP(PSPs, Network_activity)
    z_out  =  CALCULATE_READOUT_ACTIVITY(Network_
    activity)

    W = STDP(W, spiking_idx, Network_activity)
    W_out = FORCE(W_out, Network_activity)
```

## Single-trial, ISI-based Bayesian decoding

We applied a single-trial Bayesian ISI-based trial-by-trial decoding algorithm previously described[16]. In brief, using a training set composed of 90% of the data recorded trials the probability density function for observing an ISI on target/nontarget trials (or go/no-go trials) was inferred (p(target), p(non-target), p(go), and p(no-go)) via kernel density estimation with the bandwidth set by cross-validated maximum likelihood estimation. These probability density functions were used to infer the probability of a stimulus and choice from the observed ISIs, {ISI}, on a new trial taken from the remaining 10% of test trials via Bayes rule (assuming statistical independence between the ISIs observed)

$$p(stimulus|\{ISI\}) = \frac{\prod_i p(ISI_i|stimulus)\,p\,(stimulus)}{\sum_{stimulus}\prod_i p\,(ISI_i|stimulus)\,p\,(stimulus)},$$

$$p(choice|\{ISI\}) = \frac{\prod_i p\,(ISI_i|choice)\,p(choice)}{\sum_{choice}\prod_i p\,(ISI_i|choice)\,p\,(choice)}. \quad (21)$$

## Small-world path length analysis

To calculate the shortest path length shown in Supplementary Fig. 4c, we reduced our synaptic connection matrix $W$ by setting a weight threshold, culling connections whose weights fall below said threshold and using the remaining weights to define the adjacency matrix. To calculate the mean shortest path lengths of the network, we performed a breadth-first search from each neuron along its outgoing connections modified to also calculate the shortest path from a neuron to itself (either a self-edge or a cycle in the network). The mean of all

distances across all starting neurons is taken as the mean shortest path length for a given culling thresholds.

We compared these results against the same calculation performed on Watts-Strogatz small-world graphs which contain a mixture of regular local structure and random global structure. Previous work has suggested that cortical neural networks display connection statistics (such as mean path length) more consistent with a small-world network rather than one with purely random global connections[46,48]. Watts-Strogatz small-world networks vary parametrically from entirely local structure to entirely global structure via a parameter $\beta$ spanning from 0 (completely regular ring-lattice) to 1 (completely random connections between vertices). Note that this parameter is unrelated to the heterosynaptic balancing parameter $\beta$. To compare the culled adjacency matrix from our model to a Watt-Strogatz small-world network, we match the number of edges in the Watts-Strogatz graphs to the number of connections in the culled networks and generate the small-world network by first generating a bidirectional regular ring lattice with the same number of unidirectional edges as the adjacency matrix derived from our network and assign each unidirectional edge to a random endpoint with chance $\beta$.

## Inactivation experiments

Because recurrent and output connections play qualitatively different roles in network dynamics we used separate procedures to inactivate each type of connection. Full inactivation of a unit employed output and recurrent inactivation procedures simultaneously. To inactivate the output contributions of a neuron, its output weight was held at zero so that its activity made no contribution to the activity of the readout node. To inactivate a neuron's recurrent contributions, the neuron was not silenced but rather its effect on postsynaptic neurons was replaced by a Poisson process that fired with a fixed rate equivalent to that of the neuron that was being replaced. Using this approach, we could be certain that the deficits observed were not caused by an overall decrease in network synaptic current (the original unit and Poisson unit fired at the same average rate) but could instead be attributed to the removal of all spike timing information present in the unit's firing pattern.

Inactivation targeting classical/non-classical subpopulations proceeded by characterizing the firing rate modulation (see "Characterization of response profiles using a continuous measure") and inactivating in order from most to least classically/non-classically responsive unit. For example, inactivating 10% of output units during the experiment targeting the non-classically responsive subpopulation would correspond to inactivating the top 10th percentile of non-classically responsive units. Variability during the choice period was evaluated by calculating the root mean squared error between the readout node activity and correct output function on each trial.

## Synaptic motif cumulant calculations

Synaptic motif cumulants represent the extent to which a pattern of synaptic connections (e.g., a disynaptic chain) between two subpopulations (in our case, input target units, input nontarget units, output units, or inhibitory units) are present relative to chance combinations of motifs with fewer synaptic connections[50,51]. Recent work has used calculated average motif cumulants across the network, but here we extend this work by examining motif cumulants for each network unit on an individual basis to examine how these synaptic patterns relate to the response profiles of individual units.

A synaptic motif is defined as the average product of synaptic weights present in a specified pattern of connections. For example, the n-chain motif, $\mu_n^{ch}$, is the average product of synaptic weights in a n-synapse chain of unidirectional connections from one unit to

another and is calculated using the expression

$$\mu_n = \frac{\sum_{i,j} W_{ij}^n}{N^{n+1}} = \frac{\langle W^n \rangle}{N^{n-1}}. \tag{22}$$

Where $W$ is the synaptic weight matrix, $N$ is the number of network units, and $\langle \cdot \rangle$ denotes the average over all matrix elements. The two other motifs shown to be significant for network dynamics are the (m,n)-diverging and (m,n)-converging motifs, $\mu_{m,n}^{di}$ and $\mu_{m,n}^{co}$, which describe when two chains (of length $m$ and $n$) diverge from the same unit or converge onto the same unit. For 2$^{\text{nd}}$ order motifs (disynaptic) these are the only other two possible motifs beyond chain motifs and are calculated

$$\begin{aligned}
\mu_{m,n}^{di} &= \frac{\langle W^m W^{Tn} \rangle}{N^{m+n-1}}, \\
\mu_{m,n}^{co} &= \frac{\langle W^{Tm} W^n \rangle}{N^{m+n-1}}.
\end{aligned} \tag{23}$$

To decompose these motifs into motif cumulants we introduce

$$\begin{aligned}
\boldsymbol{u} &= (1,1..,1)^T / \sqrt{N}, \\
\tilde{\boldsymbol{u}} &= (1,1..,1)^T / N, \\
\boldsymbol{H} &= \boldsymbol{u}\boldsymbol{u}^T, \\
\boldsymbol{\Theta} &= \boldsymbol{I} - \boldsymbol{H}, \\
\boldsymbol{W}_\theta^n &= (\boldsymbol{W}\boldsymbol{\Theta})^{n-1}\boldsymbol{W}.
\end{aligned} \tag{24}$$

and define the n-chain, (m,n)-diverging, and (m,n)-converging motif cumulants as

$$\begin{aligned}
\kappa_n &= \frac{\tilde{\boldsymbol{u}}^T \boldsymbol{W}_\theta^n \tilde{\boldsymbol{u}}}{N^{n-1}}, \\
\kappa_{m,n}^{di} &= \frac{\tilde{\boldsymbol{u}}^T \boldsymbol{W}_\theta^m \boldsymbol{\Theta} \boldsymbol{W}_\theta^{nT} \tilde{\boldsymbol{u}}}{N^{m+n-1}}, \\
\kappa_{m,n}^{co} &= \frac{\tilde{\boldsymbol{u}}^T \boldsymbol{W}_\theta^{mT} \boldsymbol{\Theta} \boldsymbol{W}_\theta^n \tilde{\boldsymbol{u}}}{N^{m+n-1}}.
\end{aligned} \tag{25}$$

The motifs can then be decomposed in terms of the motif cumulants as

$$\begin{aligned}
\mu_n &= \sum_{\{n_1,\ldots,n_t\} \in \mathscr{C}(n)} \left( \prod_{i=1}^t \kappa_{n_i} \right) + \kappa_n, \\
\mu_{m,n}^{di} &= \sum_{\substack{\{n_1,\ldots,n_t\} \in \mathscr{C}(n) \\ \{m_1,\ldots,m_s\} \in \mathscr{C}(m)}} \left( \prod_{i=2}^t \kappa_{m_i} \right) \left( \kappa_{m_1,n_1}^{di} + \kappa_{m_1}\kappa_{n_1} \right) \left( \prod_{j=2}^s \kappa_{n_j} \right) + \kappa_{m,n}^{di}, \\
\mu_{m,n}^{co} &= \sum_{\substack{\{n_1,\ldots,n_t\} \in \mathscr{C}(n) \\ \{m_1,\ldots,m_s\} \in \mathscr{C}(m)}} \left( \prod_{i=2}^t \kappa_{m_i} \right) \left( \kappa_{m_1,n_1}^{co} + \kappa_{m_1}\kappa_{n_1} \right) \left( \prod_{j=2}^s \kappa_{n_j} \right) + \kappa_{m,n}^{co}.
\end{aligned} \tag{26}$$

where $C(n)$ represents all possible sets $\{n_1,\ldots,n_t\}$ of non-zero integers such that $\sum_{i=1}^t n_i = n$ excluding the set containing only one element $\{n\}$. The sum represents all possible ways of constructing the motif from lower order cumulants so we can interpret the final term on the right hand side as the contribution that cannot be attributed to lower order terms.

The equations above can be generalized if the network is composed of $N$ units divided into $k$ subpopulations. We specify each subpopulation $P_q$ of size $N_q$ (in our case, target input units, non-target-input units, output units, and inhibitory units) using $u_q$, a column vector of length 1 with equal entries for the components corresponding to members of the subpopulation and 0 for all other components. In other words, for subpopulation $i$ and unit $j$ the components

of vector $u_q$ are

$$u_{qj} = \begin{cases} 1/\sqrt{N_q} & \text{for } j \in P_q \\ 0 & \text{for } j \notin P_q \end{cases}. \tag{27}$$

We then specify the $N \times k$ matrix, $U$, as the concatenation of these subpopulation vectors,

$$\boldsymbol{U} = [\boldsymbol{u}_1 \ldots \boldsymbol{u}_k], \tag{28}$$

which in component notation is simply $U_{jq} = u_{qj}$. Similarly, we define $\tilde{U}$ via

$$\tilde{u}_{qj} = \begin{cases} 1/N_q & \text{for } j \in P_q \\ 0 & \text{for } j \notin P_q \end{cases}. \tag{29}$$

We now define a matrix of subpopulation motifs corresponding to the average product of synaptic weights present in specified pattern of connections where the first and last units of the motif are in subpopulation $r$ and $q$ respectively,

$$\begin{aligned}
\mu_{nqr} &= \frac{\langle W^n \rangle_{qr}}{N^{n-1}}, \\
\mu_{m,nqr}^{di} &= \frac{\langle W^m W^{Tn} \rangle_{qr}}{N^{m+n-1}}, \\
\mu_{m,nqr}^{co} &= \frac{\langle W^{Tm} W^n \rangle_{qr}}{N^{m+n-1}}.
\end{aligned} \tag{30}$$

Where $\langle \cdot \rangle_{qr}$ denotes the average over initial subpopulation $r$ and final subpopulation $q$. In matrix form we write the motifs as $k \times k$ matrices

$$\begin{aligned}
\boldsymbol{\mu}_n &= \{\mu_{nqr}\}, \\
\boldsymbol{\mu}_n^{di} &= \{\mu_{nqr}^{di}\}, \\
\boldsymbol{\mu}_n^{co} &= \{\mu_{nqr}^{co}\}.
\end{aligned} \tag{31}$$

As expected, these subpopulation motifs can be used to calculate the total motifs using a weighted average. For example, the n-chain motif is calculated from the subpopulation motifs

$$\mu_n = \sum_{q,r} \frac{N_q N_r}{N^2} \mu_{nqr} \tag{32}$$

As in the single population case, these subpopulation motifs can be broken down into a series of subpopulation motif cumulants. To calculate the subpopulation motif cumulants we follow the same process as the single population case. Defining

$$\begin{aligned}
\boldsymbol{H} &= \boldsymbol{U}\boldsymbol{U}^T, \\
\boldsymbol{\Theta} &= \boldsymbol{I}_N - \boldsymbol{H}, \\
\boldsymbol{W}_\theta^n &= (\boldsymbol{W}\boldsymbol{\Theta})^{n-1}\boldsymbol{W}.
\end{aligned} \tag{33}$$

The $k \times k$ matrix of average n-chain cumulants between each subpopulation is then calculated as

$$\boldsymbol{\kappa}_n^{ch} = \frac{1}{N^{n-1}} \tilde{\boldsymbol{U}}^T \boldsymbol{W}_\theta^n \tilde{\boldsymbol{U}} \tag{34}$$

and (m,n)-divergent and (m,n)-convergent motif cumulants are calculated

$$\kappa_{m,n}^{\mathrm{di}} = \frac{1}{N^{m+n-1}} \tilde{U}^T W_\theta^m \Theta W_\theta^{n\,T} \tilde{U},$$
$$\kappa_{m,n}^{\mathrm{co}} = \frac{1}{N^{m+n-1}} \tilde{U}^T W_\theta^{m\,T} \Theta W_\theta^{n} \tilde{U}. \tag{35}$$

We extend this previous work by using these expressions to calculate subpopulation motif cumulants for each individual network unit. Since the initial and final $\tilde{U}$ terms serve to average over the initial and final subpopulation respectively, we define the average subpopulation motif cumulants into each unit (a $N \times k$ matrix) or out of each unit (a $k \times N$ matrix) by removing the $\tilde{U}^T$ or $\tilde{U}$ term. Denoting these individual motif cumulant matrices $\hat{}$, we then have

$$\hat{\kappa}_n^{\mathrm{ch}} = \frac{1}{N^{n-1}} W_n^\theta \tilde{U},$$
$$\hat{\kappa}_n^{\mathrm{chR}} = \frac{1}{N^{n-1}} \tilde{U}^T W_n^\theta. \tag{36}$$

where $\hat{\kappa}_n^{ch}$ represents the n-chain subpopulation motif cumulants terminating in the unit of interest and $\hat{\kappa}_n^{chR}$ represents n-chain subpopulation motif cumulants originating in the unit of interest. (m,n)-divergent and (m,n)-convergent motifs are symmetric so we only require one expression for each $N \times k$ matrix.

$$\hat{\kappa}_{m,n}^{\mathrm{di}} = \frac{1}{N^{m+n-1}} W_m^\theta \Theta W_\theta^{n\,T} \tilde{U},$$
$$\hat{\kappa}_{m,n}^{\mathrm{co}} = \frac{1}{N^{m+n-1}} W_\theta^{m\,T} \Theta W_\theta^{n} \tilde{U}. \tag{37}$$

## Statistical synaptic motif model for output unit modulation

Cells which project to the output unit receive direct feedback projections, $\eta$, from the output unit, therefore we first examined the relationship between the stimulus- or choice-related firing rate modulation and the magnitude of feedback received. The output unit's activity differs systematically on "go" vs. "no-go" trials, implying that this relationship should be particularly strong for choice-related activity. This prediction is also supported by theoretical work examining the connection between connectivity and dynamics in similar networks[76]. By fitting the models

$$R_{st/ch} \sim \eta + 1 \tag{38}$$

we observed linear correlations between stimulus and choice-related responses and the strength of the feedback connections, η, both for networks including and excluding STDP with choice-related activity being the most strongly correlated ($r_{stimulus}^2 \approx 0.4, r_{choice}^2 \approx 0.8$). However, the precise relationship between output feedback and single output unit modulation varies from condition to condition (pre-STDP, post-STDP, EE only, IE only) and network to network demonstrating that this relationship is modulated by changes to the recurrent connectivity. In other words, the coefficients for this model (slope and y-intercept) were not universal and varied across and within conditions. In other words, feedback magnitude can predict which units will become the most stimulus or choice modulated within a network, but the factors that determine the precise relationship between the feedback strength and a unit's response profile are unexplained.

Our goal then was to explain the observed relationship between firing rate modulation and feedback strength using only the prevalence of individual subpopulation motif cumulants in a universal model which applies across conditions. Such a model provides a detailed understanding of the synaptic features relevant for generating

a particular response profile as it accounts for the variability using coefficients tied to each synaptic motif cumulant that apply across conditions. To accomplish this we included these motifs as linear regressors in place of the unknown coefficients from the previous model (Eq. 37). Summarizing the individual subpopulation motif cumulants of order $n_s$ (i.e., 1 is monosynaptic, 2 is disynaptic, etc.) from subpopulation $q$ (target input units, nontarget input units, output units, inhibitory units) as

$$\kappa_{n_s q} = \hat{\kappa}_{nq}^{\mathrm{ch}} + \hat{\kappa}_{nq}^{\mathrm{chR}} + \sum_{\substack{n=m=1 \\ m+n=n_s}} \left( \hat{\kappa}_{m,nq}^{\mathrm{di}} + \hat{\kappa}_{nq}^{\mathrm{co}} \right). \tag{39}$$

We can then write a model for the stimulus and choice modulation including all subpopulations $q$ and cumulants up to order $n_s$ as

$$R_{st/ch} \sim \left( \sum_{n=1}^{n_s} \sum_q \kappa_{nq} \right) \eta + \left( \sum_{n=1}^{n_s} \sum_q \kappa_{nq} \right). \tag{40}$$

Note that in this model each synaptic motif cumulant receives two coefficients: one for the interaction with the output feedback and one for feedback independent effect. All coefficients are universal in that they are fit across network conditions meaning that they quantify the contribution of each motif to stimulus and choice modulation under all observed network conditions. This model was fit using the Python statsmodels package with LASSO regularization to remove regressors with small coefficients. All calculations were done with $n_s = 2$ (monosynaptic and disyanptic motif cumulants) because higher-order models showed marginal improvement as measured by the AIC criterion. Once statistically significant coefficients were identified (p < 0.01 Bonferroni-correction for number of motif cumulants included), we refit the model with only significant motif cumulants.

To calculate the overall contribution of a particular motif $\kappa$ to either stimulus or choice modulation for a unit, $R_{st/ch}$, from Eq. 39 let $\beta_\kappa^\eta$ be the $\eta$ interaction coefficient for cumulant $\kappa$ and $\beta_\kappa$ be the independent coefficient and define the contribution from each cumulant as

$$r_{st/ch}(\kappa) = (\beta_\kappa^\eta \eta + \beta_\kappa)\kappa \tag{41}$$

such that

$$R_{st/ch} = \sum_\kappa r_{st/ch}(\kappa). \tag{42}$$

Given the definition of overall firing rate modulation, $R$, as $R^2 = R_{st}^2 + R_{ch}^2$ we can decompose it into a linear combination of motif cumulant related terms

$$\begin{aligned} R^2 &= R_{\mathrm{st}}^2 + R_{\mathrm{ch}}^2 \\ &= R_{\mathrm{st}}(\sum_\kappa r_{\mathrm{st}}(\kappa)) + R_{\mathrm{ch}}(\sum_\kappa r_{\mathrm{ch}}(\kappa)) \\ &= \sum_\kappa (R_{\mathrm{st}} r_{\mathrm{st}}(\kappa) + R_{\mathrm{ch}} r_{\mathrm{ch}}(\kappa)) \\ &\equiv \sum_\kappa r^2(\kappa) \end{aligned} \tag{43}$$

where the contribution of any given cumulant to the overall modulation squared is defined as

$$r^2(\kappa) \equiv R_{\mathrm{st}} r_{\mathrm{st}}(\kappa) + R_{\mathrm{ch}} r_{\mathrm{ch}}(\kappa). \tag{44}$$

## Spiking simulations using measured synaptic inputs

For two units, cell-attached spikes were first recorded and then after breaking into the cell, EPSCs and IPSCs were also measured for

the same neuron. For all other units, spiking activity was simulated trial-by-trial using the measured excitatory and inhibitory post-synaptic currents (E/IPSCs) via an integrate-and-fire point neuron simulation based on previously described methods[7]. The simulation relies on conductance-based dynamics

$$\tau_m \frac{dV}{dt} = (V_r - V) + g_E(t) R_m (V_E - V) + g_I(t) R_m (V_I - V) \quad (45)$$

where $V$ is the membrane voltage, $V_r$ is the resting membrane potential, and $\tau_m = R_m C_m$ is the membrane time constant. $V_{E/I}$ and $g_{E/I}(t)$ are the excitatory/inhibitory reversal potentials and time based conductances, respectively. Spiking threshold was set to $V_{th} = -40$ mV and simulated neurons were constrained with a refractory period of 5 ms.

Model parameters for individual neurons were derived using a voltage pulse of $\Delta V = 10$ mV, the membrane capacitance ($C$), input resistance ($R_i$), and series resistance ($R_s$) were calculated from the current, $I(t)$ via

$$R_s = \frac{\Delta V}{I(0)},$$
$$R_i = \frac{\Delta V}{I(\infty)}, \quad (46)$$
$$C = \frac{\tau}{R_i - R_s},$$

where $I(0)$ is the initial current on pulse onset, $I(\infty)$ is the asymptotic current, and $\tau$ is the measured exponential decay constant for $I(t)$.

PSC time courses in individual trials were fit to a standardized parametric form incorporating the maximum current ($I_{\max}$), the time of max current ($t_{\max}$), the rise time to half max ($t_{rise}$), and the fall time to half max ($t_{fall}$)

$$I(t) = I_{\max} \left( 1 - 2^{\frac{-(t - t_{\max} - t_0)}{t_{rise}}} \right) \left( 2^{\frac{-(t - t_{\max} - t_0)}{t_{fall}}} \right) \quad (47)$$

where $t_0 = t_{rise} \left( \frac{t_{fall}}{t_{rise} + 1} \right)$ is an offset requires so that $I(t_{\max}) = I_{\max}$. PSCs were then converted to the conductances, $g_{E/I}(t)$ used in the simulation by calculating the synaptic currents $I_{syn}(t) = I(t) \frac{R_{in} + R_s}{R_s}$ and converting to conductance using the holding potential, $V_h$, $g(t) = \frac{I_{syn}(t)}{V_h - V_r}$. The conductance terms used in the simulation also incorporated noise set to 10% of the average conductance values.

To generate spiking activity for an individual trial, one EPSC and one IPSC measured during stimulus presentation were randomly selected and used to calculate conductance dynamics for simulation. The process was repeated for am EPSC and IPSC taken in the 200 ms prior to stimulus onset to determine baseline activity. This process was repeated 2000 trials and the firing rate modulation during the stimulus period was calculated as the average difference between stimulus-evoked and baseline firing rate.

**Predicting in vivo firing rate modulation using motifs from spiking RNN**
To validate our spiking RNN model, we applied the statistical synaptic motif model to predict the response profiles of cortical neurons recorded from behaving animals using only measured average synaptic inputs. Because the statistical model was trained on a diversity of network structures and response profile distributions, the coefficients of our model are applicable even in cases where the overall firing statistics differ from that of the full post-STDP network.

To apply our statistical motif model to predict the stimulus-related firing rate modulation of in vivo neurons, we first fit a simplified version of our statistical model that only included monosynaptic inputs from two subpopulations (inhibition and excitation) using data from RNNs under all STDP conditions. This model predicts the stimulus modulation of an individual RNN unit from its average excitatory and inhibitory synaptic inputs. We then used this model to predict the stimulus modulation of a cell in vivo by dividing the average excitatory/inhibitory conductance values by the average number of inputs from each cell type in our model (30 for excitation and 10 for inhibition) as a proxy for that cell's average monosynaptic excitatory and inhibitory inputs. These values were then fed to our statistical model to predict the firing rate modulation.

To ensure that these predictions were non-trivial we ran two controls. The 'rescaled conductance' control preserved the average conductance values for each cell but re-ran simulations using trial-by-trial peak conductance that had beet rescaled relative to the mean conductance value for that cell (trial-by-trial peaks were brought closer to the mean by a factor of 2). This control preserved the RNN-derived predictions while altering the dynamics which produced the simulated results. The 'shuffled RNN control' randomly shuffled all unit's modulation values in our RNN before fitting the synaptic motif model to generate the coefficients predicted by chance. These coefficients were then used in place of the true coefficients to determine if the RNN-derived predictions could be explained by chance.

### Reporting summary
Further information on research design is available in the Nature Portfolio Reporting Summary linked to this article.

## Data availability
Source data are provided with this paper. Additional data that support the findings of this study are available from the corresponding authors upon request. Source data are provided with this paper.

## Code availability
Source code is available on Github[73] (https://github.com/albannalab/InsanallyAlbanna-NatComm-2024).

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

## Acknowledgements

The authors thank Larry Abbott, Tim Vogels, and David Sussillo for comments and discussions, Silvana Valtcheva for technical guidance with the whole-cell recordings, and Madeline Albanese for assisting with mouse behavioral training. This work was funded by the National Institutes of Health (grant number R00-DC015543 to M.N.I., R01-DC012557 to R.C.F., P01-NS074972 to R.C.F., and U19-NS107616 to R.C.F.), and a NARSAD Young Investigators Award to M.N.I. Shari E. Ross created artwork in Figs. 1f, 1k, 5a.

## Author contributions

M.N.I., K.K., S.F., O.L., and T.G. collected the data. B.F.A. designed the model, M.N.I, B.F.A., and J.T. performed all model simulations, K.R. and B.D. verified the model. M.N.I., B.F.A., and R.C.F. designed the study. M.N.I., B.F.A., and R.C.F. wrote the paper.

## Competing interests

The authors declare no competing interests.
