## [Peer Review File · Nature Communications]

Contributions of cortical neuron firing patterns, synaptic connectivity, and plasticity to task performanceREVIEWER COMMENTS

Reviewer #3 (Remarks to the Author):

The authors have very substantially edited the original manuscript and added new data and analysis. With this they have addressed some of my comments in a satisfactory manner. However, there are still a few important issues to solve.

1/ I have a particular concern with the abstract which echoes concern 17 of reviewer 2 (not properly addressed). One sentence is actually incomplete. Another sentence “Classically and non-classically responsive model units contribute differentially to task performance via output and recurrent connections, respectively” seem to me to convey extremely specific information related to a model that cannot be introduced in sufficient details in the abstract. The notion of classical and non-classical responders is also hard to grasp. I think the abstract should be carefully rewritten to convey the general message of the study. Wouldn't it be simpler to say that simulations in a recurrent neural network indicate that both neurons that weakly and strongly respond to sensory inputs contribute to task performance of the network. Weakly responsive neurons have a stronger impact via recurrent connectivity, while strongly responsive neurons have a larger impact via output connectivity.

2/ Overall, the terminology of classically and non-classically responsive neurons is confusing. I think it obscures the message and makes it more complex than what it is really. The authors should consider using a more explicit terminology (e.g. weakly and strongly responsive).

3/ Isn't the outcome of the study actually more intuitive, not to say evident, than what the authors suggest. The weakly responsive neurons are weakly driven by sensory inputs, and strongly responsive neurons are strongly driven. The latter provide robust information and are more strongly connected to decision outputs while the former contribute less to the output but influence indirectly the latter through recurrent connections. What have we learned from this model? Mainly that each neuron influences the network output in the artificial recurrent neuronal model. This is not really by construction but because of the plasticity mechanisms implemented, which forces a kind of balanced state. It would be much clearer if the authors simplify to recenter their claims along this line.

4/ The patch clamp recordings, although very difficult, do not bring much information to support the assumptions of the model. At best one learns that, they are not incompatible with the model. Would alternative models fail to obtain the same results? It would be much more interesting if the authors explicitly generate models that these experiments disprove.

5/ Regarding the necessity of the auditory cortex for the task, the claim has been toned down. Yet the authors should mention in the text that their conclusion is based exclusively on muscimol. The mapping of muscimol diffusion mentioned in the rebuttal (Letzkus et al.) is based on muscimol concentrations 10 times smaller than in the actual inactivation experiment (Kuchibotla 2017). It is therefore not an appropriate control. Actually it rather suggests that for Kuchibotla et al. muscimol

largely extend beyond auditory cortex. The control experiment in striatum does not provide sufficient information to be interpreted (timing of measurements?).

There are new typos in the added text:

1/ Abstract a few words are missing in one sentence. "Together both improve task performance with full network engagement."

2/ l241 representations

Reviewer #4 (Remarks to the Author):

This manuscript utilizes in vivo electrophysiological recordings from the auditory cortex of rodents performing discrimination tasks, alongside a recurrent neural network trained with STDP and FORCE to perform a similar task to make the following main claims:

- 1) The RNN trained with STDP closely matches the types and distribution of responsivity to task events as recorded in cortical neurons in vivo.
- 2) The non-responsive and responsive units both contribute to discrimination of the stimulus, but via different connections (either output connections or recurrent connections)
- 3) STDP of E>E synapses shifts the distribution of responsivity, while that of I>E synapses reduces the variance of responsivity.
- 4) Both inhibitory and excitatory plasticity improve task performance while promoting full network engagement (i.e., no inactive neurons)
- 5) The same local patterns of synaptic inputs predict responsivity of RNN units and in vivo cortical neurons.

Comments relating to how well the evidence presented backs up the scientific claims:

Comment 1 – Claim 1

My concern is how trivial it may be to generate an RNN that has no statistically significant difference in the responsivity distribution compared to the in vivo data (Figure 2e), for two reasons:

1) Extended Data Figure 5 shows that the distribution of responsiveness overtly changes with certain RNN parameters (though an important parameter, 'proportion of input units' is not analyzed). Therefore, it seems simple to find one set of parameters that has no significant difference in responsivity distribution compared to in vivo data - how sensitive is the comparison (Fig 2e) to parameter selection? Do all networks become more similar to the in vivo observation after STDP, regardless of parameters?

2) There is a very large bias in the unit recordings toward poorly responsive neurons compared to the cell attached recordings, visible when comparing Figures 1e and 1j. This likely arises from the well-known bias in unit recordings toward neurons with high firing rates (Margrie et al., 2002), which appear to be the unresponsive population. It is unlikely that distributions of responsivity from both sets of data match that of the RNN – is this the case? Given that cell attached and whole cell recordings are less biased, these are the most valid datasets for comparison (I presume the distribution from units was used for the comparison).

Overall, I am therefore not convinced that the statistically non-significant comparison between the responsivity distributions of the RNN and the in vivo data is meaningful: The in vivo distribution depends on the type of recording, and the RNN distribution depends critically on the parameters.

In addition, a heatmap of responses, rather than single examples (Figs 1d, 2i, 2d), for the NCR and CR population would be very helpful for understanding the variety of neuronal response types being described by these classifications, and how well the neurons in the RNN recapitulate these response types beyond a single index.

Comment 2 – Claim 2

I do not find this claim highly convincing given the analysis presented in Figure 3d-f. As other reviewers have mentioned, the effect size is small despite its statistical significance (especially when you take into account the y-axis range). Indeed, even at 50% inactivation, the networks on average still perform as well in terms of d' as some rats/mice (Figure 1b and g).

It would be good to see the actual dataset (i.e., points for each network), as shown for other Figures, to understand why the effect is so small but so significant, given the sample size is presumably 8 networks. It's also unclear why the analysis is repeated in the supplement but with more data points (Extended data Figure 4a-c) – why not have this as the main figure, and why does the statistical significance seemingly differ between the two analyses? Finally, actual examples of the readout node activity in each inactivation case could be helpful to understand exactly how the different inactivation cases impact the capability of the network to perform the task.

The other result used to back up this claim is that the synaptic weights for recurrent and output connections differ for NCRs and CRs (Figure 3b). Since this statistical comparison (presumably) takes place between pooled units from several networks, there is a possible nested data problem leading to exaggerated significance. Hierarchical bootstrapping (Saravanan et al., 2020) could be used to solve this.

Comment 3 - Claim 4

Full network engagement (i.e., zero inactive units) does not seem necessary to improve performance as initially suggested in the results (the d' after the inhibitory plasticity alone (Figure 3j), which acts to increase inactive units (Figure 3i), is similar to or higher than for the full STDP model (Figure 2c)). Neither is full network engagement needed to recapitulate in vivo data (many cortical neurons are silent (Jordan and Keller, 2020; Polack et al., 2013)). So, I am unconvinced there is a clear, physiologically relevant role demonstrated for the E>E plasticity.

Comment 4 - Claim 5

To make this claim more robustly, a more direct comparison could be presented for levels of I and E conductance between RNN units and those recorded in vivo (for each group: NCR and CR), as the analysis in Figure 5g is rather obscure.

Related to this, it seems that all we can conclude from 5g is that the parameters for neurons in the RNN (that relate to input current-firing rate relationship) were well chosen to accurately reflect neurons in vivo. Would coefficients from a pre-STDP network also work? The claim, in my opinion, needs restating, as the terms used, such as 'synaptic structure' and 'local pattern of inputs' suggest something more nuanced than the neuronal excitability parameters being well selected.

Finally, voltage clamp is typically highly limited in awake animals in vivo due to low input resistances and relatively high access resistances (Williams and Mitchell, 2008). The vastly different levels of conductance (and presumably R_{input}) between NCR and CR cells (Figure 5f), would suggest space clamp issues differ between them. What does this mean for results based on synaptic currents? I can see a Cesium internal was used to limit this problem as far as possible, but it is good practice to report the range of input resistances and access resistances (alongside a comparison for the two types of neuron) so that the reader can judge the quality of the voltage clamp.

General comment on cell type

Interneurons in the cortex are often untuned and have high firing rates compared to pyramidal neurons, and could match the response profile of the NCR neurons. How does the type of neuron (inhibitory or excitatory) map onto responsivity in the RNN? The fact that inactivating the NCRs leads to an increase in baseline output activity (Figure 3i) suggests that they may map onto inhibitory neurons, which could change how we interpret their contribution to performance. The same could be true of the electrophysiological recordings – one can deduce presumptive neuronal type from electrophysiological properties in whole cell recordings (Gentet et al., 2010; Pala and Petersen, 2015), and at least extract fast-spiking interneurons in unit recordings. This analysis could reveal commonalities between RNN and in vivo interneurons.

Comments relating to clarity:

The clarity and precision of the writing could be improved to increase reader comprehension. I have a non-exhaustive list of examples relating to this below:

- The use of the term ‘synaptic input’ on e.g., lines 709 and 712: ‘synaptic currents’ would be more precise.
- Is there a reason the y-axis is ‘behavioral performance’ in Figure 3d-i, and task performance in Figure 3j? The latter is probably the most accurate term for an RNN.
- As reviewer 2 previously alluded, there is still a lack of detail in methods: e.g., parameters relating to cell type, cortical layer, and recording quality should be included for electrophysiology recordings. I cannot find details of how the analysis in extended Fig. 5 occurs (e.g., before or after STDP?), or how or why normalization was used in Fig. 5f. See also my presumptions in the above text.
- The title/abstract have confusing wording: I am not sure that the phrase ‘synaptic basis of X to task performance’ makes sense.
- Lack of references for non-trivial statements, e.g., in the first paragraph, and in the paragraph starting on line 731.
- A table in the supplement detailing all statistical comparisons, sample sizes, and effect sizes, would allow better evaluation of the analyses performed. Sample sizes are particularly unclear throughout the text.

References

Gentet, L.J., Avermann, M., Matyas, F., Staiger, J.F., Petersen, C.C.H., 2010. *Neuron* 65, 422–435.

Jordan, R., Keller, G.B., 2020. *Neuron*.

Margrie, T.W., Brecht, M., Sakmann, B., 2002. *Pflugers Arch.* 444, 491–498.

Pala, A., Petersen, C.C.H., 2015. *Neuron* 85, 68–75.

Polack, P.-O., Friedman, J., Golshani, P., 2013. *Nat Neurosci* 16, 1331–1339.

<https://doi.org/10.1038/nn.3464>

Saravanan, V., Berman, G.J., Sober, S.J., 2020. *Neuron Behav Data Anal Theory* 3

Summary of revision for all referees

We thank all the reviewers for their thoughtful comments. In response to the comments and suggestions, we have expanded our results and discussion. A full list of changes and point-by-point responses to specific comments are explained here. Changes to the manuscript are highlighted in yellow. Our major changes to the manuscript include the following:

1. Revised title, abstract, and introduction to clarify meaning and motivation behind the terms “classically” and “non-classically responsive”. In response to Referee 3, we removed the terms “classically” and “non-classically” responsive in the abstract in favor of the more intuitive terms “reliable” and “irregular.” In the introduction, we now include a new paragraph motivating and defining the terms “classically” and “non-classically” responsive for the rest of the manuscript.

2. Added an additional control for the whole-cell results using only networks without STDP: This analysis was conducted in response to both Reviewers, demonstrating that STDP is required to properly infer the relationship between neuronal inputs and spiking outputs from our model.

3. Added a more fine-grained presentation of the data and results. We have included heatmaps of the evoked activity from individual neurons recorded in vivo along with the responses of individual units in our RNN model (**Extended Data Fig. 2**). We have also added panels showing the performance changes for individual RNNs during our selective inactivation experiments as well as examples of degraded readout node activity (**Extended Data Fig. 4**).

Summary of changes to revised manuscript

1. Abstract: Removed terms ‘classically’ and ‘non-classically’ responsive from the abstract and used terms more intuitive to readers before formally defining these terms in the results.

2. Introduction: Added a paragraph to the introduction to better contextualize the definitions of classically and non-classically responsive neurons and the motivation behind this study.

3. Introduction: Added citations to claims about the variability of neuronal responses.

4. Results: Clarified that inactivation results in rodents are specific to muscimol (“Diverse cortical responses measured during behavior in freely-moving rats and head-fixed mice”), and that inactivation studies can be challenging to interpret.

6. Extended Data Fig. 2a: Added single-unit activity heatmaps from RNNs and experimental data (“A spiking RNN model incorporating STDP rules captures in vivo cortical dynamics”).

7. Results: Added a series of experiments exploring the effect of input unit fraction of responsiveness (“Effect of network parameters on ensemble diversity”, **Extended Data Fig. 5c**).

8. Results: Added additional text to highlight that both classically and non-classically responsive units both contribute significantly to task performance (“Classically and non-classically responsive RNN units differentially contribute to task performance”).

9. Figure 3b,c: Employed a more nuanced statistical analysis (hierarchical bootstrapping) to account for inter-network differences (“Classically and non-classically responsive RNN units differentially contribute to task performance”).

10. Results: Clarified the meaning of ‘response variability’ in the text and **Figure 3h** (“Classically and non-classically responsive RNN units differentially contribute to task performance”).

11. Extended Data Figure 4a,c,d: Added additional panels showing the effect of inactivation on individual networks (“Classically and non-classically responsive RNN units differentially contribute to task performance”).

12. Extended Data Figure 4b: Added examples of readout node activity during inactivation experiments to demonstrate the increase of MSE and baseline shift during responsive and non-classically responsive inactivation respectively (“Classically and non-classically responsive RNN units differentially contribute to task performance”).

13. Results: We have modified how the disynaptic motif results are presented to highlight the relevance for non-classically responsive units (“Specific local synaptic patterns predict response properties of diverse units”).

14. Results: Added discussion of the importance of full network engagement (“Synaptic mechanisms shape response profile distributions and task performance”).

15. Figure 5f: Amended plot to use absolute rather than normalized conductance values (“Predicting single neuron response profiles recorded in vivo”).

16. Results: Added a control using only Pre-STDP networks to predict the firing modulation of neurons in vivo. These results demonstrate that STDP is required in our model to correctly recapitulate the input/output relationships observed in vivo (“Predicting single neuron response profiles recorded in vivo”).

17. Methods: Included input and access resistance for classically and non-classically responsive units (“Electrophysiological recordings during behavior”).

18. Discussion: Referenced Insanally et al. (eLife 2019) to support the statement: “Second, both classically and non-classically responsive units encode task information in their ISI distribution as has been observed experimentally.”

19. Discussion: Added a paragraph highlighting the other neuronal properties relevant for the biological basis of diverse response properties.

20. Entire Manuscript: Sample sizes and effect sizes added where needed, as requested by Reviewer 4.

Responses to Referee 3

1. *“I have a particular concern with the abstract which echoes concern 17 of reviewer 2 (not properly addressed). One sentence is actually incomplete. Another sentence “Classically and non-classically responsive model units contribute differentially to task performance via output and recurrent connections, respectively” seem to me to convey extremely specific information related to a model that cannot be introduced in sufficient details in the abstract. The notion of classical and non-classical responders is also hard to grasp. I think the abstract should be carefully rewritten to convey the general message of the study. Wouldn’t it be simpler to say that simulations in a recurrent neural network indicate that both neurons that weakly and strongly respond to sensory inputs contribute to task performance of the network. Weakly responsive neurons have a stronger impact via recurrent connectivity, while strongly responsive neurons have a larger impact via output connectivity.”* and:

2. *“Overall, the terminology of classically and non-classically responsive neurons is confusing. I think it obscures the message and makes it more complex than what it is really. The authors should consider using a more explicit terminology (e.g. weakly and strongly responsive).”*

Thanks, we agree that the use of the terms ‘classically responsive’ and ‘non-classically responsive’ is unclear in the abstract. We have removed the use of these technical terms in the abstract in favor of the more intuitive terms ‘reliable’ and ‘irregular’. We now define ‘classically’ and ‘non-classically’ responsive informally in the introduction, and formally our mathematical definition is in the results and methods section. In the second paragraph of the introduction, we now clarify the general problem addressed here:

“‘Classically responsive neurons’” are those that have clear trial-averaged evoked activity relative to baseline in response to stimulus or other task-related events. “Non-classically responsive” neurons are the remaining neurons that do not have these features and appear non-responsive. However, perception and behavior do not occur in a trial-averaged way but require integration and computation in real time on single trials. Thus, the contribution of classically and non-classically responsive neurons to perception and behavior is not entirely clear. In particular, the contribution of non-classically responsive cells may not be fully appreciated. Are neurons without obvious orientation tuning in V1 totally unnecessary for the perception of oriented gratings? Are neurons with place fields in the hippocampus exclusively required for spatial navigation or spatial memory or are neurons without obvious place fields important for those computations as well?”

A major result in the present manuscript is that experimentally-recorded cells which appear ‘weakly responsive’ (i.e., nominally non-responsive) are just as informative as ‘strongly responsive’ cells when their activity is quantified using spike times (specifically the ISI distribution). The is also the case for ‘weakly responsive’ model units in our network (**Fig. 2f**).

We worry that the term ‘weakly responsive’ is a bit too loaded, as well as being inaccurate. The non-classically responsive units can change their firing patterns on a trial-by-trial basis in a way that conveys information about the stimulus and/or behavioral choice; and thus that would in another sense make them ‘strongly responsive’. A more precise term might be ‘weakly rate-modulated’ but this seems to privilege event-locked rate modulation as the relevant characterization of these cells’ activity. This is a position that is in opposition to not only the decoding results of **Figure 2f**, but also the inactivation results of **Figure 3**. Thus we prefer to keep referring to these units as classically and non-classically responsive, to put them on equivalent epistemological footing. We believe that these terms- once properly defined in our manuscript in the introduction, results, and methods- better support our goal of examining the contributions of non-classically responsive cells that have been widely reported yet often neglected from analysis, but may play a critical role in neural dynamics and behavior.

3. “Isn’t the outcome of the study actually more intuitive, not to say evident, than what the authors suggest. The weakly responsive neurons are weakly driven by sensory inputs, and strongly responsive neurons are strongly driven. The latter provide robust information and are more strongly connected to decision outputs while the former contribute less to the output but influence indirectly the latter through recurrent connections. What have we learned from this model? Mainly that each neuron influences the network output in the artificial recurrent neuronal model. This is not really by construction but because of the plasticity mechanisms implemented, which forces a kind of balanced state. It would be much clearer if the authors simplify to recenter their claims along this line.”

Yes, this summary of the conclusions is correct. Major findings of our present study include: 1) each population of units (classically and non-classically responsive) contribute significantly to network performance; and 2) classically responsive units contribute primarily through output connections, while non-classically responsive units influence performance primarily through recurrent connections. **Following the recommendation of the Reviewer, we have further highlighted point #1 in our concluding paragraph on the inactivation results** (“Classically and non-classically responsive RNN units differentially contribute to task performance”) and we now state: “These inactivation experiments demonstrate that both classically and non-classically responsive units contribute significantly to task performance. Selective inactivation reveals that classically responsive units contribute more to task performance through their output projections while non-classically responsive units contribute primarily through their effect on recurrent activity.”

In addition to demonstrating the task-relevant contribution of both classically and non-classically responsive units, this manuscript also examines how these units are embedded in the network (i.e., their local synaptic architecture), how this architecture determines their response profile (i.e.,

spiking output), and (perhaps most critically) how STDP shapes this entire process, in an attempt to relate local circuitry to spiking output and behavior. We then demonstrate that the specific parameters from our network have predictive power in vivo using whole cell recordings in behaving animals (**Fig. 5**). **We reorganized and added language in the results section to clarify and highlight the interpretation of our disynaptic motif results** (“Specific local synaptic patterns predict response properties of diverse units”) to better highlight these connections.

4. *“The patch clamp recordings, although very difficult, do not bring much information to support the assumptions of the model. At best one learns that, they are not incompatible with the model. Would alternative models fail to obtain the same results? It would be much more interesting if the authors explicitly generate models that these experiments disprove.”*

We thank the reviewer for drawing attention to these results. The purpose of the whole-cell recordings was to validate predictions of our model with experimental data collected during behavior. We observed similar synaptic input and spiking output relationships as predicted by our model (**Fig. 5e,f**). This also indicates that parameters derived directly from our model are capable of successfully predicting response properties of single neurons recorded during behavior (**Fig. 5g**). To clarify these two points, in our revised manuscript we expanded the analysis shown in Figure 5. As the reviewer helpfully suggests, **we also now compare our predictions shown in Figure 5g with predictions made using only networks without STDP (Extended Data Fig. 8e).** We find that without STDP, the RNN-derived parameters systematically underpredict the firing modulation of the recorded neurons. This result demonstrates that the STDP rules included in our model can accurately predict input-output relationship observed experimentally in behaving mice

5. *“Regarding the necessity of the auditory cortex for the task, the claim has been toned down. Yet the authors should mention in the text that their conclusion is based exclusively on muscimol. The mapping of muscimol diffusion mentioned in the rebuttal (Letzkus et al.) is based on muscimol concentrations 10 times smaller than in the actual inactivation experiment (Kuchibotla 2017). It is therefore not an appropriate control. Actually it rather suggests that for Kuchibotla et al. muscimol largely extend beyond auditory cortex. The control experiment in striatum does not provide sufficient information to be interpreted (timing of measurements?).”*

Please note that we do not perform muscimol inactivations- or any experimental manipulation of activity like this- in the current manuscript. We absolutely agree that muscimol inactivation experiments have their caveats, as do all inactivation methods. It is important to consider species differences, brain size differences, volume, concentration, and even nuances in task structure. That said, the results from inactivation experiments need not be the entry point for the scientific posed in this work. For example, if inactivating a brain area did not affect a given behavior, it does not necessarily mean that activity in that region is unimportant for task

execution. While inactivation experiments have their advantages and disadvantages, the results from those experiments do not depend on the results from our manuscript, which is primarily to explore the synaptic origins of diverse response profiles in a modeling framework where this is possible.

Fundamentally, this is a complicated issue where evidence indicates that auditory cortex is important – perhaps even necessary – but we agree that these experiments can be challenging to correctly control for and interpret regardless of outcome (Slonina et al., Trends Neurosci. 2022). **We now have added a caveat to these kinds of studies in our results section** (“Diverse cortical responses measured during behavior in freely-moving rats and head-fixed mice”).

(Again, we are not including inactivations in this study ourselves, but since the reviewer is interested, here are the details of the inactivation experiments we presented in the last round of replies: For the control experiments, recordings from posterior striatum were initiated 15 minutes after the infusion of saline or muscimol in auditory cortex and continued for a duration of 30 minutes. This time frame is consistent with Kuchibhotla et al. (Nat Neurosci 2017), where behavioral trials were initiated 10-15 minutes after muscimol infusion. We found no significant change in the neural activity in posterior striatum during muscimol inactivation of auditory cortex ($p = 0.96$, Wilcoxon paired test, one way). This demonstrates that the behavioral impairment during muscimol inactivation of auditory cortex cannot be attributed to changes in posterior striatum.)

6. *“There are new typos in the added text: 1/ Abstract a few words are missing in one sentence. “Together both improve task performance with full network engagement.” 2/ 1241 representations”*

Thanks! Typos fixed. Although this may be a question for the editorial team, we believe that the sentence in the abstract – while terse – is grammatically complete: “Together, both [excitatory and inhibitory plasticity] improved task performance with full network engagement.” (We have now added a comma after “Together” to help clarify this.)

Responses to Referee 4

1. “My concern is how trivial it may be to generate an RNN that has no statistically significant difference in the responsivity distribution compared to the *in vivo* data (Figure 2e), for two reasons:

*Extended Data Figure 5 shows that the distribution of responsiveness overtly changes with certain RNN parameters (though an important parameter, ‘proportion of input units’ is not analyzed). Therefore, it seems simple to find one set of parameters that has no significant difference in responsivity distribution compared to *in vivo* data - how sensitive is the comparison (Fig 2e) to parameter selection? Do all networks become more similar to the *in vivo* observation after STDP, regardless of parameters?”*

Thanks, it’s a good suggestion to include the proportion of input units as an exploratory parameter in Extended Data Fig. 5c. As expected, varying this number also had an effect on the overall response profile distribution by changing the stimulus response of units in the network. In brief, it was non-trivial that our model would be consistent with the empirical data, nor do all sets of STDP parameters automatically make the model consistent. There is a limited parameter space of viable, performative models, and within that space, there are a limited set of models that are consistent with the response profile distributions observed *in vivo* (although the parameter values consistent with the experimental data are not entirely unique, as we discuss below). Some of the edges of the space of viable models are visible in **Extended Data Figure 1 g,h**. We now provide a more comprehensive account of the types of failures observed when STDP parameters are shifted an order of magnitude outside the current parameter regime in Reviewer Table 1.

Parameters (relative to default values)	Resulting network pathology
0.1x EE Heterosynaptic strength (β & δ)	Firing rates crash during training
0.1x IE Heterosynaptic strength (β & δ)	Low firing rates and high levels of synchrony
10x EE Homosynaptic strength	Large fluctuations in readout node activity
10x EE Heterosynaptic strength (β & δ)	Large fluctuations in readout node activity & baseline shifts in readout node activity
10x IE Homosynaptic strength	Firing rates crash during training
10x EE Heterosynaptic strength (β & δ)	Large fluctuations in readout node activity & baseline shifts in readout node activity

Reviewer Table 1: Table summarizing systematic network pathologies when STDP parameters are adjusted an order of magnitude away from default values.

One major goal of our study is to understand the connections between various STDP mechanisms and the consequent response profile distributions. If our model restricted these STDP parameters to a unique set for reasons of stability or consistency, it would be an unsuitable tool to investigate the question at hand. We could not explore the use of STDP parameters as an independent variable because the model would effectively be fixed to one allowed set of values. Conversely, it could be the case that any set of parameters leads to a stable, viable, performative model. This would be challenging for exactly the reasons stated by the reviewer: in parameter space, it would be difficult to claim that one had the ‘correct’ model for comparison to empirical data, without some kind of proof that other areas in the parameter space could not also reproduce in vivo dynamics. Given this, we argue that the ideal case is somewhere in between these two limit cases. Outside a certain range, models are no longer viable and performative (which naturally constrains the space of models that can be explored), but within the space of allowed models there is enough flexibility to change parameters in ways that affect the phenomena of interest. Crucially, there is enough flexibility to find solutions that align with the empirical evidence. Our model sits in this regime. When possible, we used previous experimental results to constrain our default values (as is the case for both homosynaptic mechanisms: Bi & Poo, J Neurosci 1998; D’amour & Froemke, Neuron 2015). Where these were not available, we began our parameter search based on previous computational work (Song & Abbott, Neuron 2001; Vogels et al., Science 2011; Zenke et al., Nat Commun 2015) and adjusted as necessary to achieve network stability and performance. In this way, finding a set of viable parameters itself was not trivial, let alone a set that would result in a distribution so closely matched to the empirical data. All viable parameter sets in our model do not match the data distribution. As we show in **Extended Data Figure 6**, changing STDP parameters leads to significant distribution shifts.

Extended Data Fig. 6b,c with experimental distribution overlaid in grey. Response profile distributions shown in Here it is evident that changes in the parameters governing each STDP mechanism can shift the distribution away/towards the empirical distribution.

Conversely, we do not claim that our parameters uniquely match the data. The manipulations in **Extended Data Fig. 6** could in principle be combined to potentially “cancel-out” their effects and remain near the observed experimental distribution (**Reviewer Fig. 1**). In most cases, this simply results in networks with response profiles that still do not match the in vivo distribution or networks that are no longer stable. That said, simultaneous manipulation of both heterosynaptic excitatory-to-excitatory mechanisms does leave the network response distribution unchanged, suggesting a low-dimensional manifold of solutions consistent with the data. This aligns with our observation that homosynaptic excitatory-to-excitatory plasticity has an extremely limited effect on the response profile distribution, and suggests an avenue for future computational and theoretical work. This does not undermine the more important observation that a solution consistent with the in vivo data exists at all within the limited space of stable models.

Reviewer Figure 1: Response profile distribution for networks where multiple heterosynaptic mechanisms were manipulated simultaneously relative to default values. Note that networks with 2x excitatory-to-excitatory heterosynaptic balancing and 2x excitatory-to-excitatory heterosynaptic enhancement have a very similar distribution to networks with default settings.

2. “There is a very large bias in the unit recordings toward poorly responsive neurons compared to the cell attached recordings, visible when comparing Figures 1e and 1j. This likely arises from the well-known bias in unit recordings toward neurons with high firing rates (Margrie et al., 2002), which appear to be the unresponsive population. It is unlikely that distributions of responsivity from both sets of data match that of the RNN – is this the case? Given that cell attached and whole cell recordings are less biased, these are the most valid datasets for comparison (I presume the distribution from units was used for the comparison). Overall, I am therefore not convinced that the statistically non-significant comparison between the responsivity distributions of the RNN and the in vivo data is meaningful: The in vivo distribution depends on the type of recording, and the RNN distribution depends critically on the parameters.”

We thank the reviewer for their thoughtful observations on the pros and cons of these various recording methods. However, the assumption that non-classically responsive cells have higher firing rates is not borne out in the data. In both rats and mice, non-classically responsive cells had lower spontaneous firing rates ($p < 0.001$, two-sided Mann-Whitney U test) so, if anything, we would expect them to be underrepresented in the rat data by the logic presented above.

That said, there are several other key differences between the single-unit recordings and cell-attached recordings which may explain the difference between their respective response profile distributions. For example, the single-unit recordings were performed in *freely-moving* rats whereas the cell-attached recordings were performed in *head-fixed* mice. The single-unit recordings were made while rats performed a frequency recognition task comprised of 7 different tones, whereas the cell-attached recordings were made while mice performed a frequency recognition task of 2 tones. Given the major differences in species, head-fixation, and task structure it's unclear whether the modulation differences are driven by recording technique or these other experimental conditions. Methods such as 2-photon imaging are arguably less biased in terms of selection as it samples high and low firing rate neurons equally. Previous work using 2-photon imaging in head-fixed mice performing the same go/no-go frequency recognition task we implemented has shown that a minority of cells are activated (enhanced) by sound during active behavior (Kuchibhotla et al, Nat. Neuro. 2017, Fig. 2h) which is consistent with the extracellular recordings performed in rats. Therefore, the single-unit rat recordings are a valid dataset to use for model comparison. The purpose of showing the cell-attached recordings in mice was to demonstrate that diverse cortical responses are universal across species, head-fixation, and behavioral conditions.

3. *“In addition, a heatmap of responses, rather than single examples (Figs 1d, 2i, 2d), for the NCR and CR population would be very helpful for understanding the variety of neuronal response types being described by these classifications, and how well the neurons in the RNN recapitulate these response types beyond a single index.”*

We thank the reviewer for their suggestion to include a population heatmap of responses. These have been included in Extended Data Figure 2a. The responses are qualitatively comparable.

4. *“I do not find this claim highly convincing given the analysis presented in Figure 3d-f. As other reviewers have mentioned, the effect size is small despite its statistical significance (especially when you take into account the y-axis range). Indeed, even at 50% inactivation, the networks on average still perform as well in terms of d' as some rats/mice (Figure 1b and g).”*

The reviewer is correct that the d' values after 50% inactivation shown in **Figure 3** are comparable to the performance of some animals, but it is not entirely valid to make a one-to-one comparison

of these metrics. In the RNN, the network is trained to produce a specific burst of activity in the readout node during the response period when the target tone is present. Its action for the purpose of d' calculations is determined by whether its output crosses a decision threshold during the response period. Even in cases where this output signal is extremely degraded, it is still possible to set a threshold which leads to relatively robust performance when the output signal is clearly degraded. This standard is clearly less complex than the task an animal has to perform where the presence of a target tone and a correct choice must be translated to downstream brain areas to make the appropriate motor movements.

It is for these reasons that we examine metrics of performance beyond d' in Figure 3. In **Figure 3h,i**, we show that inactivating these sub-populations leads to significant changes in both the readout node's response mean squared error and baseline activity, indicating that these inactivations have a dramatic effect on network dynamics even as d' performance metric changes appear more modest.

5. *"It would be good to see the actual dataset (i.e., points for each network), as shown for other Figures, to understand why the effect is so small but so significant, given the sample size is presumably 8 networks. It's also unclear why the analysis is repeated in the supplement but with more data points (Extended data Figure 4a-c) - why not have this as the main figure, and why does the statistical significance seemingly differ between the two analyses? Finally, actual examples of the readout node activity in each inactivation case could be helpful to understand exactly how the different inactivation cases impact the capability of the network to perform the task."*

We thank the reviewer for these suggestions and have added additional plots to **Extended Figure 4** showing the effect of inactivation on individual networks (**Extended Data Figure 4a-d**). While there is certainly variability in individual task performance curves, individual traces recapitulate what is summarized using the mean and SEM in **Figure 3d-f** and **Extended Data Figure 4a-d**.

The plots shown with more data points in **Extended Data Figure 4a-d** were intended to provide a more granular look at the relationship between the percent of output units inactivated. The correction used for multiple comparisons (Bonferroni correction) is quite conservative, and as such, fewer individual comparisons are deemed significant by the test. That said, there are clear inflection points at 10% inactivation in the total and output inactivation conditions (these are also visible when examining individual traces). 50% inactivation represents the condition where the entire subpopulation (classically or non-classically responsive units) are inactivated and is therefore also a key point of interest. Given these observations, we felt it appropriate to focus our analysis on these key values in the main figure and include a more detailed view in **Extended Data Figure 4**. We have also included example output node traces demonstrating the effects summarized in **Figure 3 h,i** in **Extended Data Figure 4b**.

6. *“The other result used to back up this claim is that the synaptic weights for recurrent and output connections differ for NCRs and CRs (Figure 3b). Since this statistical comparison (presumably) takes place between pooled units from several networks, there is a possible nested data problem leading to exaggerated significance. Hierarchical bootstrapping (Saravanan et al., 2020) could be used to solve this.”*

We thank the reviewer for the suggestion. The results were robust across networks, but the reviewer is right that we should account for the multi-level nature of the analysis. We have followed the approach in Saravanan et al. (2020), and found our original results still hold. The new statistical analysis used is now described in the results in the section entitled “Classically and non-classically responsive RNN units differentially contribute to task performance.”

7. *“Full network engagement (i.e., zero inactive units) does not seem necessary to improve performance as initially suggested in the results (the d' after the inhibitory plasticity alone (Figure 3j), which acts to increase inactive units (Figure 3i), is similar to or higher than for the full STDP model (Figure 2c)). Neither is full network engagement needed to recapitulate in vivo data (many cortical neurons are silent (Jordan and Keller, 2020; Polack et al., 2013)). So, I am unconvinced there is a clear, physiologically relevant role demonstrated for the $E>E$ plasticity.”*

We agree with the reviewer that it is unclear whether excitatory-to-excitatory plasticity plays a decisive role in task performance. However, it does serve to balance the effect of inhibitory-to-excitatory plasticity and maintain network engagement. While it was not significant in this study, maintaining full network engagement effectively increases the size and computational capacity of the network which may have ramifications for performance in other tasks. We now describe this in the results (“Synaptic mechanisms shape response profile distributions and task performance”).

8. *“To make this claim more robustly, a more direct comparison could be presented for levels of I and E conductance between RNN units and those recorded in vivo (for each group: NCR and CR), as the analysis in Figure 5g is rather obscure.”*

As the reviewer correctly states, our claim is not that synaptic inputs in our RNN model match those observed in vivo, but that the relationship between inputs (synaptic weights) and outputs (spiking activity) for units in our network hold for in vivo data. A direct comparison between the inputs to RNN units and neurons recorded in vivo would not support this claim – it would only establish that they have a similar distribution of inputs.

We have clarified our presentation of these results to highlight that our goal in Figure 5g is to establish that the input/output relationship observed in our RNN model is recapitulated *in vivo*.

9. *“Related to this, it seems that all we can conclude from 5g is that the parameters for neurons in the RNN (that relate to input current-firing rate relationship) were well chosen to accurately reflect neurons in vivo. Would coefficients from a pre-STDP network also work? The claim, in my opinion, needs restating, as the terms used, such as 'synaptic structure' and 'local pattern of inputs' suggest something more nuanced than the neuronal excitability parameters being well selected.”*

The suggestion to try using coefficients derived from Pre-STDP networks exclusively is interesting, and we have included it in our results (“Predicting single neuron response profiles recorded in vivo”, **Extended Data Fig. 8e**). While using coefficients derived from this Pre-STDP model makes predictions well-correlated with the measured modulation, the predictions are systematically lower than those using full STDP networks (pre-STDP MSE = 3.11 spikes/s, Full network MSE = 2.11 spikes/s, 47% decrease in MSE for full STDP vs pre-STDP only). This demonstrates that STDP is in fact required in our model to successfully capture the single neuron input/output relationship observed in vivo.

10. *“Finally, voltage clamp is typically highly limited in awake animals in vivo due to low input resistances and relatively high access resistances (Williams and Mitchell, 2008). The vastly different levels of conductance (and presumably R_{input}) between NCR and CR cells (Figure 5f), would suggest space clamp issues differ between them. What does this mean for results based on synaptic currents? I can see a Cesium internal was used to limit this problem as far as possible, but it is good practice to report the range of input resistances and access resistances (alongside a comparison for the two types of neuron) so that the reader can judge the quality of the voltage clamp.”*

We agree that we should give the full range of input resistances, separated by classically and non-classically responsive neurons. While the overall input resistance value was included in the Methods section, we now separate these values according to response type for comparison and include access resistance (end of Methods, “Electrophysiological recordings during behavior.”): “For classically responsive neurons, $R_i = 278 \pm 112 \text{ M}\Omega$ (s.d.), $R_s = 69 \pm 76 \text{ M}\Omega$ (s.d.) and for non-classically responsive neurons, $R_i = 234 \pm 84 \text{ M}\Omega$ (s.d.), $R_s = 64 \pm 54 \text{ M}\Omega$ (s.d).”

As there were no systematic differences between these values, we do not think space clamp issues differ between them. One interesting hypothesis is that some of the non-classically responsive neurons have their inputs organized differently than classically responsive neurons. In our model, units are point neurons (for obvious reasons), but of course real neurons have interesting dendritic geometries and where the inputs come in can have a huge impact on the somatic response. However, at this point, with a few exceptions (Chen et al., Nature 2011) it remains unclear where and how sensory inputs are organized in neuronal dendrites. It is an interesting idea that there could be a biological basis to the diverse response profiles related to cable filtering of more distal inputs.

We have added a paragraph to our discussion on this point.

11. *“General comment on cell type: Interneurons in the cortex are often untuned and have high firing rates compared to pyramidal neurons, and could match the response profile of the NCR neurons. How does the type of neuron (inhibitory or excitatory) map onto responsivity in the RNN? The fact that inactivating the NCRs leads to an increase in baseline output activity (Figure 3i) suggests that they may map onto inhibitory neurons, which could change how we interpret their contribution to performance. The same could be true of the electrophysiological recordings – one can deduce presumptive neuronal type from electrophysiological properties in whole cell recordings (Gentet et al., 2010; Pala and Petersen, 2015), and at least extract fast-spiking interneurons in unit recordings. This analysis could reveal commonalities between RNN and in vivo interneurons.”*

This is an interesting point. As mentioned in comment #2 above, non-classically responsive units do not have systematically higher firing rates than classically responsive cells. In **Extended Data Fig. 2b**, we show that both excitatory and inhibitory units are equally likely to be classically or non-classically responsive (excluding excitatory input units, which are by definition classically responsive).

12. *“The use of the term ‘synaptic input’ on e.g., lines 709 and 712: ‘synaptic currents’ would be more precise.”*

Good idea, we’ve made these changes.

13. *“Is there a reason the y-axis is ‘behavioral performance’ in Figure 3d-i, and task performance in Figure 3j? The latter is probably the most accurate term for an RNN.”*

Thanks, we fixed the axis label.

14. *“As reviewer 2 previously alluded, there is still a lack of detail in methods: e.g., parameters relating to cell type, cortical layer, and recording quality should be included for electrophysiology recordings. I cannot find details of how the analysis in EDF 5 occurs (e.g., before or after STDP?), or how or why normalization was used in Fig. 5f. See also my presumptions in the above text.”*

Thanks for pointing this out. We added additional details about the methods for the single-unit rat recordings, including cortical layer and recording quality metrics (under “Electrophysiological recordings during behavior”). That information is already present for the cell- attached recordings. We have clarified in the text and caption to **Extended Data Figure 5** that all networks shown include STDP parameters (“Effect of network on ensemble diversity”).

Normalization was conducted to highlight the difference between excitatory and inhibitory conductances but the reviewer is right to point out that this may add unnecessary confusion for the reader. **We have amended this panel to use the conductance measured to avoid confusion for the reader (Fig. 5f).**

15. The title/abstract have confusing wording: I am not sure that the phrase 'synaptic basis of X to task performance' makes sense.

Thanks, we adjusted the title to read "Synaptic basis of diverse cortical neuron responses and contributions to task performance."

16. "Lack of references for non-trivial statements, e.g., in the first paragraph, and in the paragraph starting on line 731."

We have added citations to the first paragraph of the introduction and second paragraph of the discussion.

17. "A table in the supplement detailing all statistical comparisons, sample sizes, and effect sizes, would allow better evaluation of the analyses performed. Sample sizes are particularly unclear throughout the text."

Thanks and sorry that we didn't include this before. We believe these data are best included in legends and the results so that the reader can see them when needed. **We have added sample and effect sizes throughout the manuscript in both the text and figure captions.**

REVIEWER COMMENTS

Reviewer #3 (Remarks to the Author):

The authors have addressed adequately several of my issues, but a few clarifications are still needed.

1/ 2/ The authors have made a substantial effort to clarify what they meant with non-classically and classically responsive neurons, at least in the rebuttal letter. Mentioning the decoding analysis of the previous study (Elife 2019), indicating that there is information in NCR / weakly rate-modulated neurons is key for understanding the logic of the paper, and must appear in the definition of the two neuron classes. However, these precisions are not reflected in the manuscript.

- Based on the operational definition in the methods classically responsive neurons are neurons responding by a firing rate modulation above a particular threshold. Non-classically responsive are all other neurons. Based on this, rate-modulated and weakly rate-modulated seems a more explicit terminology.

- The paragraph aimed at clarifying the definition of CR and NCR neurons is only half convincing. The authors should definitely keep the first two sentences but remove or drastically reduce the rest of the paragraph which is out of scope. My suggestion;

“

‘Classically responsive neurons’ / ‘rate-modulated neurons’ are those that have clear trial-averaged evoked activity relative to baseline in response to stimulus or other task-related events. Non-classically responsive neurons’ / ‘weakly rate modulated neurons’ are the remaining neurons. As shown previously (Elife 2019), despite the weak rate modulation, the remaining neurons encode substantial information about the stimulus or task-related event in the relative timing of their spikes. It is therefore important to determine whether these neurons contribute to the computations performed by the auditory cortex during task performance.

“

3&4/ These points were addressed adequately.

5/ The description of reversible inactivation experiments is still extremely partial. The authors should cite other works using optogenetics, which contradict the results of muscimol experiments. This is all the more important that one of the co-authors of the present study has released optogenetic results that indicate clearly that auditory cortex is dispensable for performing the type

of pure tone discriminations used here
<https://www.abstractsonline.com/pp8/#!/10892/presentation/36805>.

Letzkus et al. 2011, cited by the authors, should be updated with a more recent paper of the Letzkus lab indicating that primary auditory cortex is not involved in pure tone discrimination (Dalmay et al., Neuron 2019, published together with Ceballo et al. 2019).

The consensus view. There is a broad literature describing the effect of lesions in mice, gerbils, cats and non-human primate, demonstrating without ambiguity that auditory cortex is dispensable for pure tone discrimination as summarized in ref [38]. The extent to which auditory cortex is involved when preserved is still debated due to unresolved contradictions between muscimol and optogenetic experiments.

I suggest the following wording to be more balanced:

“

Perturbation experiments in rodents have revealed that the contribution of auditory cortex to perceptual decision making depends on details of the task design and difficulty [38]. Muscimol inactivation studies suggests that the auditory cortex is required for this task [7,16,39,40]. However, lesion and optogenetic inactivation studies suggest on the contrary that auditory cortex is dispensable for similar tasks (REF). The difference between these loss of function experiments may still require appropriate controls for unambiguous interpretation [38].

“

Reviewer #4 (Remarks to the Author):

The authors have addressed many of my comments, but two concerns remain given the author responses:

1) A concern I still have is that no the mismatch between RNN and the mouse cell attached dataset is not addressed. Strikingly, based on the medians and IQRs in the text, while STDP brings the

responsivity distribution of the RNN closer to that of the rat unit dataset (Figure 2g), STDP pushes the distribution further away from that of the mouse cell attached dataset.

I find the explanation in the author responses for the RNN only being compared to the rat dataset unconvincing (head-fixation could increase CRs but it is not clear why head-fixation would make this dataset less comparable to the RNN, while presenting fewer tone frequencies presumably should bias the mouse recordings to fewer CRs, not more). If the difference is real and not caused by methodological differences, given that both rats and mice are performing a similar task with high accuracy, this suggests that the proportion of NCRs in cortex doesn't matter much for performing tasks - in which case, what is the significance of STDP altering the RNN responsivity distribution at all?

Therefore, the claim that 'the RNN trained with STDP closely approximated the types and distribution of responsivity as recorded in cortical neurons in vivo' is only true for one dataset, and it is unclear what the significance is that STDP changes the distribution. My suggestion is that the issue is discussed alongside a comparison for rat and mouse responsivity distributions. In addition, the existing statements and figures comparing in vivo data and RNN responsivity distributions should clearly refer to the rat unit recording dataset only.

2) Another concern arises from the response of the authors, where it is stated that NCR neurons have lower baseline firing rates on average than CR neurons (though the data is not presented). If this is the case, why do the NCR and CR examples presented indicate the opposite trend (Figure 1g and 1i)? If real-data NCRs on average have lower baseline firing rates than CRs, does this mean that this does not match the RNN, where average firing rates are higher for NCRs (Figure 2d examples, Figure 3c, line 312)? I would suggest adding the comparison of baseline firing rates for the in vivo data as has been done for the RNN and selecting more representative examples of NCRs and CRs for the in vivo data in Figure 1. It is important to understand which aspects of in vivo responsivity the RNN captures, and which aspects are different.

Minor points

1) The title has been changed, though I still find the wording quite unclear in terms of what it is trying to convey – the contributions of what to task performance?

2) There is an inconsistency in the text, given the claim of a close approximation between RNN and data responsivity distributions: Line 205: 'we found the relative fractions of classically and non-classically responsive units were comparable to experimental measurements (Extended Data Fig.2b-d; ~40-50% non-classically responsive, 50-60% classically responsive)'. Compare this to line 123: 'Most single-units recorded during behavior were generally non-classically responsive, with a minority of cells having more 'classical' responses, e.g., to tone presentation.'

Summary of revision for all referees

We thank the reviewers for their continued thoughtful feedback. In response to the comments and suggestions, we have made changes to our title, figures, introduction, results, and discussion. A full list of changes and point-by-point responses to specific comments are explained here. Changes to the manuscript are highlighted in **yellow**. Major changes to the manuscript include:

- 1. Title:** As per the reviewer suggestion, we changed the title to “Contributions of cortical neuron firing patterns, synaptic connectivity, and plasticity to task performance”, to make more it more obvious that these relations are the topic of our manuscript.
- 2. Introduction:** Clarified our definitions of non-classically and classically responsive neurons.
- 3. Figure 1:** Chose examples in **Fig. 1d,i** more representative of baseline firing statistics.
- 4. Results:** Updated text on ratios of non-classically vs classically responsive units. (“Diverse cortical responses measured during behavior in freely-moving rats and head-fixed mice”)
- 5. Extended Data Figure 1:** Added a comparison of baseline firing rates of non-classically responsive neurons and responsive neurons for both rats and mice.
- 6. Results:** Included comparison of baseline firing rates for both mice and rats. (“Diverse cortical responses measured during behavior in freely-moving rats and head-fixed mice”, third paragraph.)
- 7. Results:** Included language to clarify the significance of response profile distributions in both rats and mice (“Diverse cortical responses measured during behavior in freely-moving rats and head-fixed mice”, third paragraph)
- 8. Figure 2:** Updated examples and alignment of visual representation of PSTHs in **Figure 2d**.
- 9. Figure 2:** Included both rat and mouse modulation CDFs in **Figure 2e**.
- 10. Results:** Added specific comparison between rat and mouse experimental data (“A spiking RNN model incorporating STDP rules captures in vivo cortical dynamics”, fourth paragraph).
- 11. Figure 3:** Added baseline firing rate comparison to **Figure 3c**.
- 12. Results:** Added additional text to reference new **Figure 3c** and discuss differences between relative firing rates of non-classically and classically responsive units in the model and *in vivo*. (“Classically and non-classically responsive RNN units differentially contribute to task performance”, first paragraph.)
- 13. Discussion:** Expanded discussion comparing model response profiles and experimental data.
- 14. Discussion:** Expanded our discussion of the role of auditory cortex in pure tone tasks and potential caveats of particular inactivation methods (second paragraph).

Responses to Referee 3

We thank the referee for appreciating our efforts to revise our manuscript and their enthusiasm for our results.

1. *“The authors have made a substantial effort to clarify what they meant with non-classically and classically responsive neurons, at least in the rebuttal letter. Mentioning the decoding analysis of the previous study (Elife 2019), indicating that there is information in NCR / weakly rate-modulated neurons is key for understanding the logic of the paper, and must appear in the definition of the two neuron classes. However, these precisions are not reflected in the manuscript.*

- Based on the operational definition in the methods classically responsive neurons are neurons responding by a firing rate modulation above a particular threshold. Non-classically responsive are all other neurons. Based on this, rate-modulated and weakly rate-modulated seems a more explicit terminology.

- The paragraph aimed at clarifying the definition of CR and NCR neurons is only half convincing. The authors should definitely keep the first two sentences but remove or drastically reduce the rest of the paragraph which is out of scope. My suggestion:

‘Classically responsive neurons’ / ‘rate-modulated neurons’ are those that have clear trial-averaged evoked activity relative to baseline in response to stimulus or other task-related events. Non-classically responsive neurons’ / ‘weakly rate modulated neurons’ are the remaining neurons. As shown previously (Elife 2019), despite the weak rate modulation, the remaining neurons encode substantial information about the stimulus or task-related event in the relative timing of their spikes. It is therefore important to determine whether these neurons contribute to the computations performed by the auditory cortex during task performance.”

We agree with the reviewer’s suggestions to clarify the definition of classically and non-classically responsive neurons **and have modified the second introductory paragraph significantly along the lines that the reviewer suggests**. The paragraph now reads:

“Classically responsive neurons” (or “strongly rate-modulated neurons”) are those that have clear trial-averaged evoked activity relative to baseline in response to stimulus or other task-related events. “Non-classically responsive neurons” (or “weakly rate-modulated neurons”) are the remaining neurons that do not have these features and appear non-responsive. As shown previously [16], despite little to no rate modulation, non-classically responsive neurons encode substantial information about stimulus or choice behavioral variables in the relative timing of their spikes. Thus, it is important to determine whether these neurons contribute to the computations underlying task performance.”

2. *“The description of reversible inactivation experiments is still extremely partial. The authors should cite other works using optogenetics, which contradict the results of muscimol experiments. This is all the more important that one of the co-authors of the present study has released optogenetic results that indicate clearly that auditory cortex is dispensable for performing the type of pure tone discriminations used here:*

<https://www.abstractsonline.com/pp8/#!/10892/presentation/36805>.

Letzkus et al. 2011, cited by the authors, should be updated with a more recent paper of the Letzkus lab indicating that primary auditory cortex is not involved in pure tone discrimination (Dalmy et al., Neuron 2019, published together with Ceballo et al. 2019).

The consensus view. There is a broad literature describing the effect of lesions in mice, gerbils, cats and non-human primate, demonstrating without ambiguity that auditory cortex is dispensable for pure tone discrimination as summarized in ref [38]. The extent to which auditory cortex is involved when preserved is still debated due to unresolved contradictions between muscimol and optogenetic experiments. I suggest the following wording to be more balanced:

Perturbation experiments in rodents have revealed that the contribution of auditory cortex to perceptual decision making depends on details of the task design and difficulty [38]. Muscimol inactivation studies suggests that the auditory cortex is required for this task [7,16,39,40]. However, lesion and optogenetic inactivation studies suggest on the contrary that auditory cortex is dispensable for similar tasks (REF). The difference between these loss of function experiments may still require appropriate controls for unambiguous interpretation [38].”

This is an important point, thanks for your helpful suggestions and references. **We have updated the citations accordingly in the second paragraph of the discussion section.** We also appreciate your helpful suggestions on how to tackle this nuanced issue and have modified the discussion accordingly. **The second paragraph of the discussion now reads:**

“Perturbation experiments in rodents have revealed that the contribution of auditory cortex to perceptual decision making depends on details of the task design and difficulty [38]. Muscimol inactivation studies suggest that auditory cortex is required for auditory stimulus classification tasks such as the one considered here [7,16,39,58]. However, lesion and optogenetic inactivation studies that auditory cortex is not required for similar tasks [38,59,60]. The difference between these types of loss of function experiments may still require appropriate controls for unambiguous interpretation [38].”

Responses to Referee 4

We thank the referee for their helpful suggestions and continued interest in our manuscript.

1. *“The authors have addressed many of my comments, but two concerns remain given the author responses: A concern I still have is that no the mismatch between RNN and the mouse cell attached dataset is not addressed. Strikingly, based on the medians and IQRs in the text, while STDP brings the responsivity distribution of the RNN closer to that of the rat unit dataset (Figure 2g), STDP pushes the distribution further away from that of the mouse cell attached dataset.*

I find the explanation in the author responses for the RNN only being compared to the rat dataset unconvincing (head-fixation could increase CRs but it is not clear why head-fixation would make this dataset less comparable to the RNN, while presenting fewer tone frequencies presumably should bias the mouse recordings to fewer CRs, not more). If the difference is real and not caused by methodological differences, given that both rats and mice are performing a similar task with high accuracy, this suggests that the proportion of NCRs in cortex doesn't matter much for performing tasks - in which case, what is the significance of STDP altering the RNN responsivity distribution at all?

Therefore, the claim that ‘the RNN trained with STDP closely approximated the types and distribution of responsivity as recorded in cortical neurons in vivo’ is only true for one dataset, and it is unclear what the significance is that STDP changes the distribution. My suggestion is that the issue is discussed alongside a comparison for rat and mouse responsivity distributions. In addition, the existing statements and figures comparing in vivo data and RNN responsivity distributions should clearly refer to the rat unit recording dataset only.

Thanks for these suggestions, which we have now incorporated into our revised manuscript. Our study was guided by two main questions: 1) Given that non-classically responsive cells have been widely reported in the literature for decades, do they contribute to task performance? 2) Given that we know responses are shaped by synaptic plasticity mechanisms, how does STDP affect the distribution of classically and non-classically responsive distributions?

The reviewer is correct that the distribution of our spiking RNN with STDP corresponds closely with one empirical response distribution (rat) it differs from the other (mouse). However, **it is important to note that our network architecture reproduces a range of response profiles as observed *in vivo* (in both rats and mice)** and STDP parameters are capable of systematically altering this distribution. The observation that the spiking RNN response distribution agrees with the rat distribution testifies to the ability of our model to capture the response statistics of one empirical dataset even though only a subset of model parameters allow for stable network performance (as discussed in our previous reply).

To clarify this point in the manuscript, we have added the distribution of mouse cortical neuronal responses to Figure 2e, and added additional text to clarify that our model qualitatively

reproduces a wide range of response profiles as seen in vivo and recapitulates the rat (but not the mouse) response profile distribution in detail. (“A spiking RNN model incorporating STDP rules captures in vivo cortical dynamics”, fourth paragraph.)

2. *“Another concern arises from the response of the authors, where it is stated that NCR neurons have lower baseline firing rates on average than CR neurons (though the data is not presented). If this is the case, why do the NCR and CR examples presented indicate the opposite trend (Figure 1g and 1i)? If real-data NCRs on average have lower baseline firing rates than CRs, does this mean that this does not match the RNN, where average firing rates are higher for NCRs (Figure 2d examples, Figure 3c, line 312)? I would suggest adding the comparison of baseline firing rates for the in vivo data as has been done for the RNN and selecting more representative examples of NCRs and CRs for the in vivo data in Figure 1. It is important to understand which aspects of in vivo responsivity the RNN captures, and which aspects are different.”*

Thanks, this is a useful suggestion. **We have modified the manuscript and figures to make comparisons between the activity of units in the spiking RNN model and the empirical neural activity easier for readers to compare and characterize.** To this end we have: 1) **added a new Extended Data Figure 1** comparing the baseline firing rates for classically and non-classically responsive neurons in both rats and mice; 2) selected example PSTHs in **Figure 1d,i** that are more representative of typical baseline firing statistics for both classically and non-classically responsive neurons; 3) updated the PSTHs for units in the spiking RNN to make them more representative and bring them in line with the visual style of the experimental PSTHs in **Figure 1**; and 4) included a comparison between the relative baseline firing rates of classically to non-classically responsive cells for the spiking RNN model (**Fig. 3c**).

We have amended the first paragraph of “Classically and non-classically responsive RNN units differentially contribute to task performance” to read:

“This result, coupled with the observation that non-classically responsive units generally had higher baseline and full-trial firing rates (Fig. 3c, median baseline NCR = 17.6 spikes/s vs. CR = 15.4 spikes/s, full-trial NCR = 17.8 spikes/s vs. CR = 15.9 spikes/s, both comparisons $p < 10^{-5}$, $N = 8$ networks, $n = 4,800$ units, Hierarchical Bootstrapping Test) suggests non-classically responsive units may play a privileged role in generating task-related dynamics through their effect on recurrent network activity. Notably, in vivo average baseline firing rates for non-classically responsive neurons were lower than classically responsive neurons in both rats and mice (Extended Data Fig. 1) this difference may result from the fact that default network parameters prevented neurons with very low average firing rates from emerging.”

3. *“The title has been changed, though I still find the wording quite unclear in terms of what it is trying to convey – the contributions of what to task performance?”*

We agree, and we have now modified the title to make the connection between task performance and neural responses, synaptic connectivity, and plasticity more grammatically transparent. The title is now: “Contributions of cortical neuron firing patterns, synaptic connectivity, and plasticity to task performance”.

4. *“There is an inconsistency in the text, given the claim of a close approximation between RNN and data responsivity distributions: Line 205: ‘we found the relative fractions of classically and non-classically responsive units were comparable to experimental measurements (Extended Data Fig.2b-d; ~40-50% non-classically responsive, 50-60% classically responsive)’. Compare this to line 123: ‘Most single-units recorded during behavior were generally non-classically responsive, with a minority of cells having more ‘classical’ responses, e.g., to tone presentation.’”*

Thank you for catching this. We have corrected the second paragraph of “Diverse cortical responses measured during behavior in freely-moving rats and head-fixed mice” to read: “Many single-units recorded during behavior were non-classically responsive, with the remainder of cells having more ‘classical’ responses, e.g., to tone presentation.”

REVIEWERS' COMMENTS

Reviewer #3 (Remarks to the Author):

The authors have adequately addressed my remaining comments.

Reviewer #4 (Remarks to the Author):

The authors have addressed all of my remaining concerns with the changes they have made. I still find the significance of STDP altering the responsivity profile of the population questionable, since it seems like the proportion of NCRs has little to do with task performance in the experimental data. In addition, given the NCRs are typically low firing rate neurons in vivo, it is unclear whether they would contribute to task performance in the same way as those in the RNN, which are typically higher firing rate (than CRs). However, these remaining questions are difficult to address, and should not influence acceptance of the manuscript.

Summary of revision for all referees

We thank the reviewers for their thoughtful feedback throughout this process. We have made minor changes to the manuscript to align with Nature's editorial policy:

- **Figure 3 g-i** were moved to **Supplemental Figure 5 c-e** to comply with Figure sizing requirements and caption length requirements.
- Sample sizes, statistical procedures, and p-values were further clarified where appropriate.
- The captions for **Figure 2** and **Figure 3** were shortened to comply with length requirements.
- **Figure 4** was rearranged to comply with size requirements.

Response to Reviewer #3

The authors have adequately addressed my remaining comments.

Thank you for your actionable and substantive feedback throughout this process.

Response to Reviewer #4

The authors have addressed all of my remaining concerns with the changes they have made. I still find the significance of STDP altering the responsivity profile of the population questionable, since it seems like the proportion of NCRs has little to do with task performance in the experimental data. In addition, given the NCRs are typically low firing rate neurons in vivo, it is unclear whether they would contribute to task performance in the same way as those in the RNN, which are typically higher firing rate (than CRs). However, these remaining questions are difficult to address, and should not influence acceptance of the manuscript.

We thank the reviewer for their deep engagement with the manuscript. We agree that there is still important work to be done to characterize non-classically responsive units in vivo and assess their contributions to behavior directly. We are encouraged that our modeling approach will provide a framework for future research on the significance of diverse neural response profiles.